# SPATIA: Multimodal Generation and Prediction of Spatial Cell Phenotypes

Zhenglun Kong [*1]  Mufan Qiu [*2]  John Boesen [1]  Xiang Lin [1]  Sukwon Yun [2]  Tianlong Chen [2]  Manolis Kellis [3]
Marinka Zitnik [1]

## Abstract

Understanding how cellular morphology, gene expression, and spatial context jointly shape tissue function is a central challenge in biology. Image-based spatial transcriptomics technologies now provide high-resolution measurements of cell images and gene expression profiles, but existing methods typically analyze these modalities in isolation or at limited resolution. We address the problem by introducing SPATIA, a multi-level generative and predictive model that learns unified, spatially aware representations by fusing morphology, gene expression, and spatial context from the cell to the tissue level. SPATIA also incorporates a spatially conditioned generative framework with confidence-aware OT reweighting and morphology-profile alignment for modeling target-state morphology distributions. Specifically, we propose a confidence-aware flow matching objective that reweights weak optimal-transport pairs based on uncertainty. We further apply morphology-profile alignment to encourage biologically meaningful image generation, enabling the modeling of microenvironment-dependent phenotypic transitions. We assembled a multi-scale dataset consisting of 25.9 million cell-gene pairs across 17 tissues. We benchmark SPATIA against 18 models across 12 tasks, spanning categories such as phenotype generation, annotation, clustering, gene imputation, and cross-modal prediction. SPATIA achieves improved performance over state-of-the-art models, improving generative fidelity by 8% and predictive accuracy by up to 3%.

[1]Department of Biomedical Informatics, Harvard Medical School, Boston, MA, USA [2]Department of Computer Science, University of North Carolina, Chapel Hill, NC, USA [3]Department of Computer Science, Massachusetts Institute of Technology, Cambridge, MA, USA. Correspondence to: Marinka Zitnik <marinka@hms.harvard.edu>.

*Proceedings of the 43rd International Conference on Machine Learning*, Seoul, South Korea. PMLR 306, 2026. Copyright 2026 by the author(s).

## 1. Introduction

Understanding the interplay between cellular morphology, gene expression, and spatial organization is essential for modeling tissue function and cell states in health and disease (Szałata et al., 2024; Stirling et al., 2021). Image-based spatial transcriptomic (ST) technologies enable high-resolution profiling of gene expression in intact tissue, along with morphology derived from microscopy images (Ståhl et al., 2016; Chen et al., 2015; Janesick et al., 2023). However, existing approaches often analyze morphology and gene expression separately, limiting their ability to learn representations of cellular phenotypes within spatial context.

The central challenge is to learn unified representations that (i) capture the joint structure between image and gene modalities (Chelebian et al., 2025; Min et al., 2024), (ii) preserve spatial dependencies at the cell level (Birk et al., 2025; Wen et al., 2023), and (iii) generalize across levels from local niches to whole-slide tissue context (Schaar et al., 2024). Naive fusion strategies, such as simple concatenation, fail to capture the nonlinear, context-dependent relationships essential in spatial omics, where cellular identity is shaped by tissue architecture.

Existing models fall short in integrating these dimensions at cell-level resolution. Single-cell models typically ignore morphology (Cui et al., 2023; Kalfon et al., 2025) or focus on spot-level correlations (Tian et al., 2024; Wang et al., 2025a; Wen et al., 2023; Schaar et al., 2024; Li et al., 2025). Pathology models excel at whole-slide analysis but disregard molecular information (Chen et al., 2022; 2024b). Vision-language models rely on textual supervision and often struggle with compositional reasoning and spatial grounding (Huang et al., 2023; Lu et al., 2024; Ding et al., 2024; Lu et al., 2023). Even recent multimodal ST models operate only at patch resolution, lacking cell granularity (Lin et al., 2024; Chen et al., 2024a).

These limitations span three dimensions. First, current methods fail to capture the full range of morphological and expression variation at *cell-level resolution*, which is essential for defining cell identity. Second, they do not model *cross-level spatial interactions*, ignoring how local niches and global tissue organization govern biological processes. Third, they cannot accurately predict *microenvironment-*

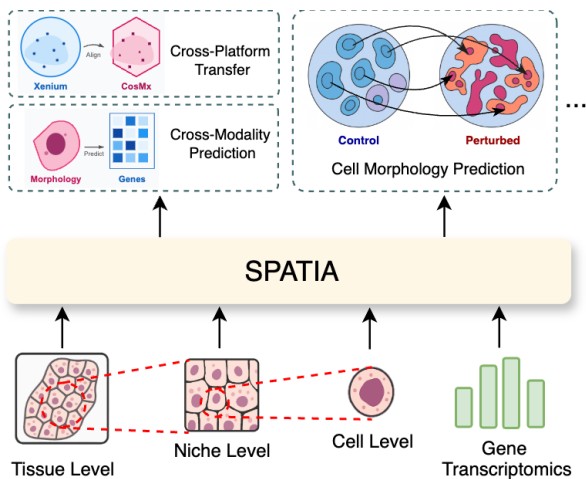

*Figure 1.* SPATIA is a multimodal generative and predictive model of spatial cell phenotypes.

*dependent morphological changes* under perturbations. Unlike generic image synthesis, modeling these effects requires generative approaches that respect both the intrinsic cell state and the extrinsic spatial niche.

**Present Work.** We introduce SPATIA[1], a multi-level model for generative and predictive modeling of spatial cell phenotypes (Fig. 1). SPATIA integrates cell morphology, gene expression, and spatial coordinates within a unified model. The model consists of three components. At the *cell level*, we fuse image-derived morphological tokens and transcriptomic embeddings via cross-attention. At the *niche level*, SPATIA groups neighboring cells into spatial patches (e.g., 256×256 pixels) and applies a transformer to model local cell-cell interactions. At the *tissue level*, a global transformer aggregates niche representations to capture long-range dependencies across the full slide. Each instance links morphology and gene expression at matched spatial levels, enabling fine-grained multimodal representation learning.

More importantly, SPATIA introduces a spatially conditioned image-to-image generation module designed to predict morphological outcomes of perturbations without paired pre-post data. We construct weak supervision pairs between control and perturbed cells within spatially adjacent or niche-consistent regions, using entropy-regularized Optimal Transport (OT) in gene expression space to align distributions. To address the inherent noise in weak pairing, we propose a *confidence-aware* flow matching framework that explicitly reweights flow trajectories based on OT coupling uncertainty. We further enhance generation with *morphology-profile alignment* to ensure generated cells respect biological

---

[1]SPATIA website: https://zitniklab.hms.harvard.edu/SPATIA/; code: https://github.com/mims-harvard/SPATIA/tree/main

feature distributions and *condition-contrastive* regularization to maximize state identifiability. This design allows SPATIA to faithfully simulate microenvironment-dependent changes, such as DCIS-to-invasive progression and immune-cold to immune-hot remodeling.

SPATIA is trained on MIST (Multi-scale dataset for Image-based Spatial Transcriptomics), a newly assembled multi-level dataset of image-based spatial transcriptomics. MIST contains 25.9 million cell-gene pairs, 2 million niche-gene pairs, and 20,000 tissue-gene pairs from 74 sources, spanning 17 tissues, 60 donors, and four platforms (Fig. 3ABC). Across 12 tasks, SPATIA outperforms 18 existing models, achieving an 8% improvement in generative fidelity and up to 3% gains in predictive benchmarks.

Our main contributions are:

- **Hierarchical multi-level architecture:** We introduce SPATIA, a multimodal model that integrates morphology, gene expression, and spatial coordinates. SPATIA represents cells, local niches, and whole-tissue context through hierarchical attention, allowing the model to capture spatial structure across biological scales.
- **Spatially conditioned generative modeling:** We develop a conditional flow-matching module for morphology generation under perturbations. The module conditions on intrinsic cell state and extrinsic spatial niche, and uses optimal transport to construct weak control-target pairs without paired pre-post perturbation data.
- **Morphology-profile alignment:** We introduce a morphology-profile alignment objective that constrains generated cells in an interpretable phenotypic feature space. This objective encourages generated cells to match target-state morphology distributions, rather than only image-level appearance.
- **Large-scale benchmarking:** We validate SPATIA on MIST, a curated dataset of 25.9M cells from 74 sources. Across 12 predictive and generative tasks, SPATIA outperforms 18 existing models, improving generative fidelity by 8% and predictive performance by up to 3%.

**Conflict of Interest Disclosure.** The authors declare no financial conflicts of interest or other substantive conflicts that could reasonably be perceived to influence the work presented in this paper.

## 2. Related Work

**Spatial Transcriptomics Models.** Recent models include scGPT-spatial (Wang et al., 2025a), which continually pretrains scGPT on multiple platforms of spatial data; CellPLM (Wen et al., 2023), pretrained on spatially resolved transcriptomic data to encode inter-cell relations; CellSymphony (Acosta et al., 2025), which achieves accurate cell type annotation and uncovers distinct microenvironmental niches.

SpaGCN (Hu et al., 2021), STAligner (Zhou et al., 2023), and SpaOTsc (Cang & Nie, 2020) integrate spatial transcriptomics with histology, but primarily at spot- or patch-level rather than true cell multimodality. Additionally, most methods operate at spot-level resolution (Vicari et al., 2024; Tian et al., 2024; Yang et al., 2025; Wang et al., 2024), lack cell-level granularity, and neglect integration of high-resolution histology or full-slide spatial context.

**Computational Pathology Models.** Vision-only models, such as HIPT (Chen et al., 2022) and UNI (Chen et al., 2024b), utilize hierarchical and self-supervised ViT pretraining on gigapixel WSIs. Vision-language approaches such as CONCH (Lu et al., 2024) and TITAN (Ding et al., 2024) employ contrastive and generative alignment with captions and reports to enable retrieval and report generation. Multimodal image-omic models such as ST-Align (Lin et al., 2024), STimage-1K4M (Chen et al., 2024a), HEST-1k (Jaume et al., 2024) integrate spatial transcriptomics and morphology for gene expression inference and cell mapping. However, existing models are also constrained to spot-level resolution and do not capture cell-level granularity, which is crucial for dissecting cellular heterogeneity and microenvironmental interactions. Vision-only models lack explicit neighborhood or multi-level tissue context, whereas vision-language models heavily depend on textual annotations, which can vary in quality.

**Generative Models.** Diffusion-based (Ho et al., 2020; Dhariwal & Nichol, 2021) and flow-matching-based generative models (Lipman et al., 2022) are powerful frameworks that transform noise into structured outputs, enabling high-fidelity and conditional synthesis. In the biomedical domain, such models have been increasingly applied to capture the complexity of cellular systems. For example, cellular morphology painting (Navidi et al., 2025), gene expression prediction (Huang et al., 2025b;a; Zhu et al., 2025), Simulating Cellular Morphology Changes (Zhang et al., 2025; Wang et al., 2025c; Palma et al., 2025), and Modeling Microenvironment Trajectories (Sakalyan et al.). Optimal transport (Cuturi, 2013; Tong et al., 2023) is also widely used for computational biology (Klein et al., 2025) More related works are provided in the Appendix D.

# 3. SPATIA Model

## 3.1. Problem Formulation and Notation

SPATIA learns multi-level representations from image-based spatial transcriptomics by integrating (i) single-cell morphology and gene expression and (ii) spatial context across niches and tissue. We consider a spatial transcriptomics dataset $\mathcal{D} = \{(x_i, \mathbf{g}_i, \mathbf{s}_i)\}_{i=1}^N$, where $x_i$ represents the high-resolution crop of cell morphology, $\mathbf{g}_i$ denotes the gene expression vector, and $\mathbf{s}_i$ indicates spatial coordinates. We

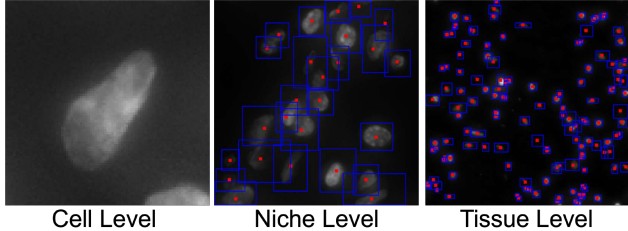

| Cell Level | Niche Level | Tissue Level |

*Figure 2.* Example of the three levels of MIST dataset.

define the *niche* as the local spatial neighborhood of cell $i$. SPATIA addresses two coupled objectives:

**Unified Representation Learning.** We aim to learn a fusion encoder $\mathcal{F}$ that maps a cell and its spatial context to a unified embedding $\mathbf{z}_i = \mathcal{F}(x_i, \mathbf{g}_i, \mathbf{s}_i)$. This embedding integrates intrinsic features (cell image and gene tokens) with extrinsic context (niche and tissue-level dependencies) to support downstream biomedical tasks.

**Spatially Conditioned Generation.** We treat perturbation modeling as a conditional generation problem. Given a control cell state $(x_{ctrl}, \mathbf{g}_{ctrl})$ and a transition type $\tau$ (e.g., biological perturbation), we aim to generate the target morphology $x_{tgt}$. As paired $(x_{ctrl}, x_{tgt})$ observations are unavailable in destructive spatial transcriptomics, we rely on *weak supervision*. We formulate the pairing via Entropy-Regularized Optimal Transport (OT) on gene expression, yielding a coupling matrix $\mathbf{P}^*$. We then model the morphological transition using Flow Matching, learning a velocity field $v_\theta$ that transports flow latents $\ell$ from control to target distributions, conditioned on the learned embeddings $\mathbf{z}_{ctrl}$ and transition signatures.

## 3.2. Unified Single-Cell Representation Learning

We propose a hierarchical framework to learn a unified embedding $\mathbf{z}_i$ for each cell $i$, integrating its intrinsic state with multi-scale spatial context (Fig. 2). We derive level-specific embeddings: $\mathbf{z}_{cell}$, $\mathbf{z}_{niche}$ (local), and $\mathbf{z}_{tissue}$ (global), and fuse them via a final projection: $\mathbf{z}_i = \mathcal{F}_{\text{fusion}}(\mathbf{z}_{cell}, \mathbf{z}_{niche}, \mathbf{z}_{tissue})$. This unified representation supports diverse downstream tasks, ranging from cell type annotation and spatial identification to the spatially conditioned morphology generation described in Sec. 3.3.

For each cell, we learn a unified embedding that integrates morphology and gene expression. We encode the cropped cell image $x$ with a ViT-based encoder to obtain visual cell level tokens: $\mathbf{X}_{cell} = E_{\text{cell}}(x)$. In parallel, we encode the gene $\mathbf{g}$ using the pretrained single-cell encoder as backbone (Kalfon et al., 2025), yielding gene tokens $\mathbf{X}_{gene} = E_{\text{gene}}(\mathbf{g})$. We then fuse modalities via cross-attention with visual tokens as queries:

$$\mathbf{z}_{cell} = \text{Attn}(Q = \mathbf{X}_{cell}, \ K = \mathbf{X}_{gene}, \ V = \mathbf{X}_{gene}), \quad (1)$$

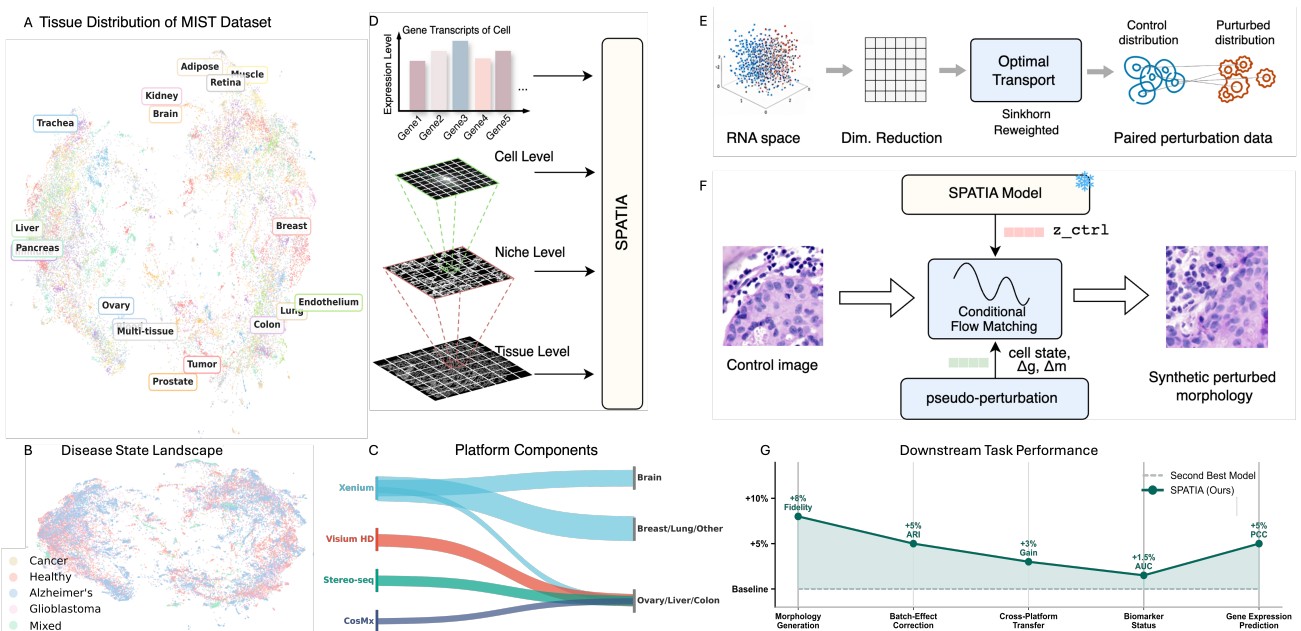

*Figure 3.* A) Tissue distribution of our MIST dataset. B) A landscape showcasing the variety of disease states in MIST. C) MIST contains four platforms containing different tissue and organ types. D) Overview of SPATIA. E) Processing controltarget pairs with optimal transport. F) Our conditional contrastive flow matching approach for predicting cell morphology. G) Downstream task performance gain compared to existing models.

where $\mathbf{z}_{cell}$ represents a cell embedding that aligns morphology of this cell with the gene expression within this cell. We refine cell embeddings by modeling spatial organization at two levels: niches and tissue (slide). Each niche is a spatial region containing a set of neighboring cells (Fig. 2). For each niche region, we pool cell embeddings to obtain a niche query vector: $\mathbf{q}_{niche} = \text{Pool}(\{\mathbf{z}_{cell}\})$, and encode the niche image: $\mathbf{X}_{niche} = E_{\text{niche}}(x_{niche})$. Cross-attention yields a niche embedding:

$$\mathbf{z}_{niche} = \text{Attn}(Q = \mathbf{q}_{niche}, \ K = \mathbf{X}_{niche}, \ V = \mathbf{X}_{niche}). \quad (2)$$

Note that the aggregation used at the niche level is only intended to construct a coarse regional context. It is not used to replace cell-level morphology–expression modeling. All cell-resolved tasks retain the original single-cell inputs, and multimodal fusion is performed through cross-attention rather than linear aggregation, allowing SPATIA to preserve cell identity while using niche and tissue signals as context.

Similarly, for *tissue* level, we aggregate niche embeddings and positional encodings, capturing long-range dependencies across regions. The resulting $\mathbf{z}_{niche}$ embeddings are contextualized to the *tissue* and serve as spatially-aware context for downstream tasks.

Cell-level modeling is necessary because several target tasks, including annotation and clustering, require preserving single-cell identity and paired morphology-expression correspondence. Niche and tissue representations are therefore used as contextual signals that complement the cell-level representation.

### 3.3. Spatially Conditioned Morphology Generation

Cell morphology under perturbation is shaped by both intrinsic cell state and the surrounding microenvironment, yet paired before/after observations are unavailable in destructive spatial transcriptomics. SPATIA therefore learns *control-to-target* morphology generation from weakly paired cells/niches and conditions generation on (i) an instance-specific control context and (ii) a transition-specific perturbation descriptor that encodes molecular and morphology-profile shifts.

**Weak Pair Construction.** We construct weak control–target pairs denoted as $(x_{\text{ctrl}}, \mathbf{g}_{\text{ctrl}}; x_{\text{tgt}}, \mathbf{g}_{\text{tgt}})$, where $x_{\text{ctrl}}$ and $x_{\text{tgt}}$ denote the cell images; $\mathbf{g}_{\text{ctrl}}$ and $\mathbf{g}_{\text{tgt}}$ are the corresponding gene expression vectors. These pairs are formed between biologically related cells (same lineage or niche-consistent regions). For example, low-malignancy vs. invasive epithelial cells or immune-cold vs. immune-hot T cells. Pairing is formulated as entropy-regularized optimal transport in reduced PCA expression space and solved by Sinkhorn (Cuturi, 2013; Tong et al., 2023). Crucially, OT is performed in gene expression space rather than in image space to avoid trivial morphology matching; spatial proximity constraints further reduce mismatches due to tissue heterogeneity. Detailed pairing method is described in Appendix C.2.

**Confidence-Aware OT Reweighting.** OT-based matches are imperfect; incorrect pairings introduce noise into the endpoint displacement supervision used in latent flow matching, defined as $\ell_{\text{tgt}} - \ell_{\text{ctrl}}$ for control–target pairs $(x_{\text{ctrl}}, x_{\text{tgt}})$. We propose to use OT coupling strength as a reliability signal. For each sample $x_{\text{ctrl}}$, we define confidence score

$$c(x_{\text{ctrl}}) = \max_{x_{\text{tgt}}} \mathbf{P}^*(x_{\text{ctrl}}, x_{\text{tgt}}), \tag{3}$$

where $\mathbf{P}^*$ is the entropy-regularized OT coupling matrix. We convert this confidence score into a normalized training weight:

$$w(x_{\text{ctrl}}) = \frac{c(x_{\text{ctrl}})^\gamma}{\mathbb{E}_{x_{\text{ctrl}}}\left[c(x_{\text{ctrl}})^\gamma\right]}, \tag{4}$$

where $\gamma$ controls how strongly uncertain pairs are downweighted. The normalization keeps the expected loss scale stable across batches, so reweighting changes the relative contribution of weak pairs rather than the overall optimization magnitude. We apply $w(x_{\text{ctrl}})$ only to the OT-positive flow supervision term, allowing high-confidence OT pairs to contribute more to the estimated velocity field while reducing the influence of ambiguous matches. This stabilizes conditional flow learning under weak supervision while retaining the diversity of weakly paired samples.

**Spatial Perturbation Embedding.** Our goal is to condition morphology generation not only on instance-specific control state but also on transition level state that summarizes the typical molecular and morphological shift associated with a transition type $\tau$ (state $A \to B$ under a cell/niche context). We define gene-expression shift signature and morphology shift signature as expectations over the paired set:

$$\begin{cases} \Delta\mathbf{g} = \mathbb{E}\big[\mathbf{g}_{tgt} - \mathbf{g}_{ctrl}\big], \\ \Delta\mathbf{m} = \mathbb{E}\big[\mathrm{M}(x_{tgt}) - \mathrm{M}(x_{ctrl})\big] \end{cases} \tag{5}$$

where $\mathrm{M}(\cdot)$ denotes morphology features (e.g., CellProfiler). $\Delta\mathbf{m}$ is computed once per transition from training pairs and does not require per-sample target morphology features as input at inference, avoiding target leakage while making the condition explicitly morphological.

We project the two signatures to low-dimensional summaries $(\widetilde{\Delta\mathbf{g}}, \widetilde{\Delta\mathbf{m}})$ and then fuse them with a learned transition token embedding $\mathbf{z}_{trans} = \mathrm{Embed}(\tau)$:

$$\mathbf{z}_{pert} = \mathrm{MLP}_{pert}\Big([\mathbf{z}_{trans}; \widetilde{\Delta\mathbf{g}}; \widetilde{\Delta\mathbf{m}}]\Big). \tag{6}$$

where $\mathbf{z}_{pert}$ is a morphology-aware perturbation condition that complements gene-based OT pairing by explicitly encoding the expected morphology-profile shift for each transition type. $\mathrm{MLP}_{pert}$ denotes a learnable projection. Details of transition design, OT pairing/QC, and signature computation are provided in Appendix C.2.

**Inference-time Conditioning.** Importantly, $\Delta m$ and $\Delta g$ are transition-level statistics computed once from weakly paired training samples, rather than per-sample target features. At inference time, SPATIA requires only the input control cell image, its control representation $z_{cond}$, and the precomputed transition descriptor $z_{pert}$. No target morphology image, target gene expression profile, or target Cell-Profiler feature is used for an individual test sample. Thus, SPATIA performs conditional target-state distribution generation for unseen control cells within an observed transition family, rather than reconstructing a measured target cell.

**Flow Matching for Control-to-Target Generation.** Given a control image and its OT-matched target, we encode endpoints $\ell_{ctrl}, \ell_{tgt}$, and define the ground-truth velocity $\boldsymbol{u} = \ell_{tgt} - \ell_{ctrl}$. We sample the linear bridge:

$$\ell_\lambda = (1-\lambda)\ell_{ctrl} + \lambda\ell_{tgt}, \quad \lambda \sim \mathcal{U}(0,1). \tag{7}$$

We define $\mathbf{z}_{ctrl}$ as the instance-specific control context extracted from SPATIA (embedding). We form the final condition embedding by fusing $\mathbf{z}_{ctrl}$ and $\mathbf{z}_{pert}$.

Next, we train a conditional velocity field $v_\theta(\ell_\lambda, \lambda \mid \mathbf{z}_{cond})$ using confidence-reweighted flow matching:

$$\mathcal{L}_{FM}^w = \mathbb{E}_\lambda\Big[w\left\|v_\theta(\ell_\lambda, \lambda \mid \mathbf{z}_{cond}) - \boldsymbol{u}\right\|_2^2\Big]. \tag{8}$$

Here the expectation is taken over paired samples (and $\lambda$), and $w$ downweights uncertain OT matches.

**Condition-Contrastive Regularization.** Weak pairing can cause overlap between transition-conditioned distributions, making it difficult for the model to distinguish different $\tau$. To encourage *condition identifiability* without pulling toward incorrect endpoints, we construct an incorrect transition condition by replacing $\tau$ with $\tau^- \neq \tau$ while keeping the same control context $\mathbf{z}_{cond}^- = \mathrm{MLP}_{cond}([\mathbf{z}_{ctrl}; \mathbf{z}_{pert-}])$.

We then enforce that the true condition better explains the ground-truth velocity than the incorrect one:

$$\mathcal{L}_{cond} = \mathbb{E}_\lambda\Big[\|v_\theta(\ell_\lambda, \lambda \mid \mathbf{z}_{cond}) - \boldsymbol{u}\|_2^2 \\ - \|v_\theta(\ell_\lambda, \lambda \mid \mathbf{z}_{cond}^-) - \boldsymbol{u}\|_2^2\Big]_+. \tag{9}$$

where $[\cdot]_+$ prevents unbounded decrease and implements a margin-style preference for correct conditioning.

### 3.4. Training

**SPATIA Training.** Before downstream adaptation, SPATIA is trained with reconstruction-based self-supervision on paired morphology–expression data. The unified cell embedding is optimized to reconstruct both the input cell image and the corresponding gene-expression profile, encouraging

it to retain complementary morphological and molecular information. We additionally enforce cross-modal consistency between image-derived and expression-derived representations so that the learned embedding remains informative when one modality is noisier or partially missing. This pretraining stage provides the initialization used for downstream prediction and conditional morphology generation.

**Morphology-Profile Alignment.** Weak OT pairing provides noisy supervision, so the generator may fail to match the target morphology distribution even with correct transition conditioning. We propose a morphology-profile alignment loss in the same feature space used for evaluation.

Let $\phi(\cdot)$ be a differentiable morphology encoder pretrained to regress CellProfiler features; we freeze $\phi$ during training to avoid representation drift. For each mini-batch, we define two empirical distributions $\mathcal{D}$ in morphology feature space $\mathcal{D}_{gen} = \{\phi(\hat{x}_{tgt})\}$ and $\mathcal{D}_{real} = \{\phi(x_{tgt})\}$, and align them using sliced Wasserstein distance: $\mathcal{L}_{morph} = \text{SWD}(\mathcal{D}_{gen}, \mathcal{D}_{real})$. We use a distributional objective since strict pixel-level pairing is unreliable under weak OT matches $\mathcal{L} = \mathcal{L}_{FM}^{w} + \rho \mathcal{L}_{cond} + \lambda_{morph}\mathcal{L}_{morph}$, where $\mathcal{L}_{morph}$ enforces diagnostic fidelity by matching generated and real target morphology-profile distributions.

# 4. Experiments

## 4.1. Datasets and Experimental Setup

**MIST Datasets.** MIST (Multi-scale dataset for Image-based Spatial Transcriptomics) dataset is assembled from 74 sources (Janesick et al., 2023; Ren et al., 2025; Gabitto et al., 2024), spanning 17 tissues, 60 donors, and four platforms. It comprises three nested scales: MIST-C (25.9M single cellgene pairs), MIST-N (2M nichegene pairs), and MIST-T (20K tissuegene entries), enabling morphologytranscriptomics mapping at cellular, regional, and whole-slide levels for multimodal learning across diverse biological contexts. We load full-resolution tissue images ($0.2125\mu$m/px), compute maximum-intensity projections over z, and normalize the resulting 2D images to 8-bit $[0, 255]$.

Using cell boundary files, we extract individual cells by cropping the minimal square region that fully contains each cell. Each crop is resized with a slide-level global scale and padded to a fixed $256 \times 256$ resolution, preserving biologically meaningful variation in cell size while preventing neighboring-cell pixels from being included in an image paired with a different expression vector. Each MIST-C example consists of this uint8 image patch paired with the per-cell transcript vector of a single gene, serialized into LMDB for efficient training. For MIST-N, each slide is tiled into non-overlapping $256 \times 256$ px niches; cells are assigned to their containing patch, and gene vectors are pooled within each niche. Each entry thus contains a regional image

patch and its aggregated gene profile. MIST-T summarizes each slide using its set of niche embeddings and positional metadata at $1024 \times 1024$ resolution, supporting tissue-level tasks such as global composition prediction and cross-slide transfer. Full dataset statistics are provided in Appendix E.

**Baselines.** We benchmark SPATIA against competitive models, including CellFlux (Zhang et al., 2025), GeneFlow (Wang et al., 2025b) and MorphDiff (Wang et al., 2025c) for cell morphology prediction; UNI (Chen et al., 2024b), GigaPath (Xu et al., 2024), Hibou (Nechaev et al., 2024), PLIP (Huang et al., 2023), CONCH (Lu et al., 2023), CTransPath (Wang et al., 2022) and H-Optimus-0 (Saillard et al., 2024) for biomarker status prediction and gene expression prediction; as well as single-cell models: scFoundation (Hao et al., 2024a), Nicheformer (Schaar et al., 2024), Geneformer (Theodoris et al., 2023), scGPT (Cui et al., 2023), CellTypist (Domínguez Conde et al., 2022), UCE (Rosen et al., 2023), scBERT (Yang et al., 2022), and CellPLM (Wen et al., 2023) for cell annotation & clustering. While many baselines are designed primarily for either prediction or generation, SPATIA is built to jointly support conditional morphology generation and prediction.

**Experimental Setup and Implementation.** For generative evaluation, we assess SPATIA on control-to-target generation of cell morphology. For predictive evaluation, we benchmark SPATIA across four groups of tasks: cell annotation, clustering, gene expression prediction, and biomarker status prediction. We mainly followed the downstream evaluation settings from (Wen et al., 2023) and (Jaume et al., 2024). We conduct 3 individual runs for tasks. Evaluations use donor-disjoint 70/10/20 train/validation/test splits, so morphology, expression, and spatial context from the same donor never appear across different splits.

We use four NVIDIA H100 GPUs for 25K steps with the hierarchical batching strategy keeping memory usage bounded. This makes the framework practical for large spatial transcriptomics datasets while preserving cell-level resolution. Detailed training settings and model configurations are provided in Appendix A.

## 4.2. Control-to-Target Generation of Cell Morphology

SPATIA models cross-modal transitions to enable morphology generation under biological state changes. We define each generation task as a biologically constrained population-level transition between annotated source and target cell states, rather than as a paired before/after observation of the same cell. We consider two transition axes: 1) tumor progression from ductal carcinoma in situ (DCIS) to invasive carcinoma, instantiated by luminal epithelial to EMT-associated epithelial transitions; and 2) immune remodeling from immune-cold to immune-hot microenvironments, instantiated by T-cell activation and angiogenesis-

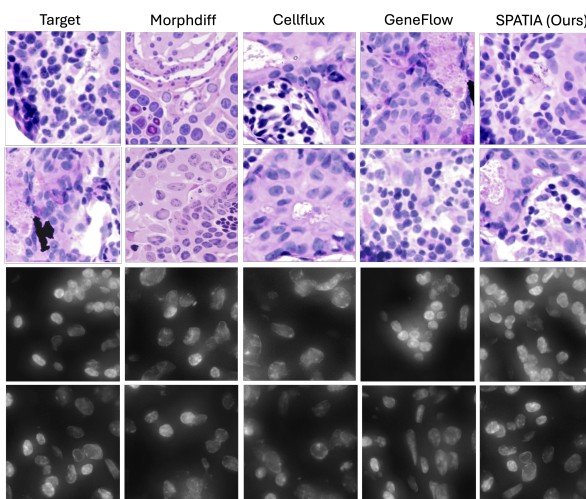

*Figure 4.* Qualitative image analysis of generated cell morphology change images with the target image. Target denotes real cells sampled from the target state distribution.

*Table 1.* Conditional morphology generation. Progressive addition of SPATIA components improves both image fidelity and biological morphology correctness.

| Method | Image fidelity | | Morphology correctness | |
|---|---|---|---|---|
| | FID ↓ | KID ↓ | Wass. Corr. ↑ | KS ↑ |
| CellFlux | 64.1 ± 1.15 | 2.31 ± 0.09 | 0.83 ± 0.026 | 0.57± 0.030 |
| MorphDiff | 70.5 ± 1.52 | 2.52± 0.11 | 0.81 ± 0.037 | 0.54 ± 0.034 |
| GeneFlow | 62.4 ± 1.28 | 2.20± 0.08 | 0.87± 0.025 | 0.58± 0.020 |
| SPATIA (base) | 59.5 ± 1.16 | 2.09 ± 0.12 | 0.90 ± 0.028 | 0.61± 0.023 |
| + Reweight | 59.1 ± 1.19 | 2.06 ± 0.15 | 0.91± 0.031 | 0.62± 0.022 |
| + Morph. Loss | **58.5± 1.05** | **2.01± 0.07** | **0.94± 0.024** | **0.65± 0.025** |

associated endothelial activation. To avoid biologically implausible weak pairs, we restrict control and target cells to compatible lineages, require at least 50 cells per state, and prefer OT matches between spatially similar niches.

Evaluation proceeds along two axes. Image fidelity is assessed using Frechet Inception Distance (FID) (Heusel et al., 2017) and Kernel Inception Distance (KID) (Bińkowski et al., 2018), providing standard measures of generative realism. Morphological correctness is assessed using CellProfiler-derived features (Carpenter et al., 2006).

For each feature, generated and real distributions in the target state are compared using statistical distances such as the Kolmogorov–Smirnov (KS) statistic (Massey Jr, 1951) and the Wasserstein distance (Panaretos & Zemel, 2019). This dual evaluation ensures that generated images are not only visually realistic but also biologically faithful. Models are trained on the same donor-disjoint 70/10/20 split. We match conditioning according to each works native interface: CellFlux uses control-image plus perturbation conditioning, MorphDiff uses perturbed transcriptomic conditioning, and GeneFlow uses gene-expression embeddings.

*Table 2.* Cross-platform evaluation on cell clustering. Higher NMI/ARI indicate better biological structure preservation.

| Model | Xenium | | CosMx | |
|---|---|---|---|---|
| | ARI ↑ | NMI ↑ | ARI ↑ | NMI ↑ |
| scGPT | 0.730±0.033 | 0.678±0.013 | 0.507±0.048 | 0.472±0.013 |
| scFoundation | 0.727±0.006 | 0.754±0.006 | 0.530±0.039 | 0.560±0.007 |
| Nicheformer | 0.256±0.026 | 0.345±0.002 | 0.064±0.007 | 0.089±0.001 |
| UCE | 0.618±0.006 | 0.718±0.006 | 0.516±0.015 | 0.555±0.010 |
| SPATIA | **0.735±0.007** | **0.806±0.004** | **0.542±0.043** | 0.490±0.009 |

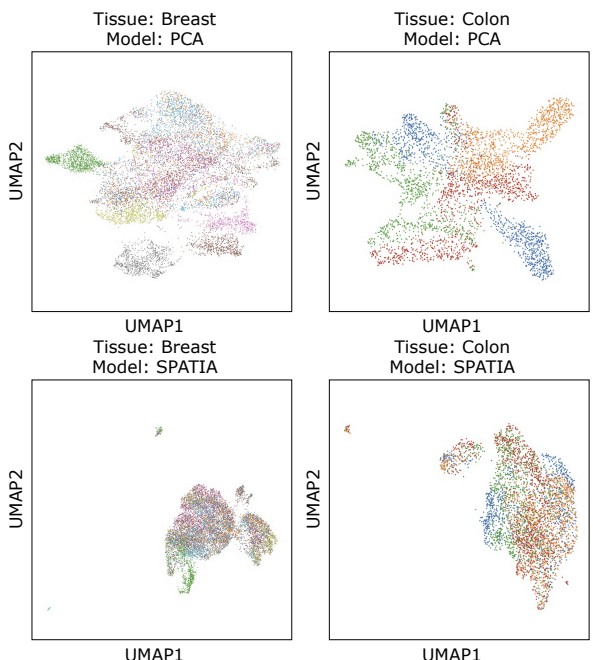

*Figure 5.* Batch-effect mitigation. Different colors represent different datasets or sample sources.

Tab. 1 highlights both the image fidelity and biological validity of our generative framework. Compared to GeneFlow, CellFlux, and MorphDiff, our method achieves lower FID and KID scores, indicating a more realistic synthesis of images (mean over 3 runs). Higher Wasserstein correlations and KS statistics demonstrate that generated morphologies more faithfully reproduce the distribution of CellProfiler-derived features in the target states. We show visualization comparisons of generated samples in Fig. 4.

### 4.3. Cross-Platform & Batch Effect Evaluation

We evaluate whether SPATIA learns platform-agnostic representations by benchmarking cell clustering on two distinct spatial transcriptomics platforms: Xenium and CosMx (both include DAPI-stained data). Pretrained embeddings are frozen and evaluated using clustering consistency metrics (ARI/NMI). We compare against single-cell models, including scFoundation (Hao et al., 2024a), scGPT, Nicheformer (Schaar et al., 2024), and UCE (Rosen et al.,

*Table 3.* Receptor–status prediction evaluation on BCNB

| Model | ER | | PR | | HER2 | |
|---|---|---|---|---|---|---|
| | AUC | Bal.acc. | AUC | Bal.acc. | AUC | Bal.acc. |
| GigaPath | 0.841 | 0.765 | 0.803 | 0.696 | 0.721 | 0.635 |
| Hibou | 0.832 | 0.754 | 0.801 | 0.694 | 0.705 | 0.630 |
| CLIP | 0.652 | 0.537 | 0.618 | 0.502 | 0.514 | 0.438 |
| PLIP | 0.712 | 0.603 | 0.695 | 0.587 | 0.611 | 0.524 |
| CONCH | 0.881 | 0.745 | 0.810 | 0.698 | 0.715 | 0.624 |
| UNI | 0.891 | 0.775 | 0.820 | 0.712 | 0.732 | 0.641 |
| SPATIA | **0.902** | **0.785** | **0.825** | **0.730** | **0.744** | **0.643** |

*Table 4.* Cell annotation and clustering results.

| Method | Annotation | | Method | Clustering | |
|---|---|---|---|---|---|
| | F1 (↑) | Prec. (↑) | | ARI (↑) | NMI (↑) |
| scGPT | 0.703 | 0.729 | PCA | 0.843 | 0.812 |
| CellPLM | 0.709 | 0.702 | CellPLM | 0.867 | 0.823 |
| scBERT | 0.599 | 0.604 | scGPT | 0.856 | 0.828 |
| CellTypist | 0.667 | 0.693 | Geneformer | 0.461 | 0.586 |
| SPATIA | **0.725** | **0.734** | SPATIA | **0.870** | **0.831** |

2023). We select the best Leiden resolution from the same candidate set, and report the final performance as mean ± std over 5 seeds. All methods were re-evaluated under the same updated method for fair comparison.

As shown in Tab. 2, SPATIA achieves better clustering performance on both platforms, demonstrating that its multimodal representations better preserve biological structure across different acquisition technologies. Fig. 5 qualitatively compares the batch-effect mitigation capabilities of SPATIA across tissues aggregated from multiple sources, where our model demonstrates superior integration performance. We attribute this robustness to the hierarchical classification loss and adversarial training strategies employed during the scPRINT pre-training; notably, we retained this adversarial training objective during fine-tuning to explicitly enforce the learning of batch-invariant representations.

### 4.4. Biomarker Status Prediction

We evaluate SPATIA on receptor-status prediction (ER, PR, HER2) using WSIs from the BCNB dataset (Tab. 3). Following (Chen et al., 2022), histology patches are embedded with a pretrained pathology encoder, while gene expression features are encoded using a lightweight 3-layer MLP. The modality-specific embeddings are aligned via a contrastive (InfoNCE) objective by fine-tuning the image encoder and training the expression encoder from scratch.

Despite not being designed as a task-specific classifier, SPATIA matches or exceeds specialized pathology models optimized for supervised prediction across all three biomarkers. Gains over strong models such as UNI are consistent across markers rather than concentrated on a single task, demonstrating that the learned spatial representations support both generative modeling and downstream predictive tasks. Results are averaged over 3 runs, with standard deviations below 0.01 for AUC and 0.012 for balanced accuracy.

### 4.5. Cell Annotation & Clustering

We follow the settings in (Wen et al., 2023) and use Multiple Sclerosis (MS) dataset (Schirmer et al., 2019) to evaluate

cell annotation performance and sc-RNAseq data (Li et al., 2020). Results are shown in Tab. 4. We report F1 and Precision scores for the annotation task; ARI and NMI scores for the clustering task. These results highlight SPATIA's ability in supervised and unsupervised single-cell analysis tasks compared to existing methods.

Although predictive improvements over state-of-the-art pathology and single-cell models are modest, it is important to note that these models are optimized *purely* for prediction tasks (Wang et al., 2025a; Chen et al., 2024b; Lu et al., 2023; Xu et al., 2024), which lack generative capabilities. SPATIA's strong performance on predictive tasks and state-of-the-art performance on generative tasks (Tab. 1) shows that it is possible to train a single model jointly for both predictive and generative modeling of spatial cell phenotypes.

### 4.6. Gene Expression Prediction from Images

We use HEST-Bench (Chen et al., 2022) to evaluate SPATIA for gene expression prediction task. Fig. 6 reports Pearson correlation coefficients (PCC) for the top 50 highly variable genes on five cancer cohorts (IDC, PAAD, SKCM, COAD, LUAD). We train a regression model to map model-specific patch embeddings to the log1p-normalized expression of the top 50 highly variable genes. We use XGBoost (Chen & Guestrin, 2016) regression model with 100 estimators and a maximum depth of 3, while embeddings are frozen. These consistent gains across diverse tissue types demonstrate that SPATIA yields embeddings that more accurately capture gene-image relationships than existing single or dual modal architectures. Across tasks, SPATIA maintains competitive or superior predictive performance while uniquely enabling controllable morphology generation under biological transitions. This highlights that modeling cross-modal generative structure yields representations that generalize across prediction tasks without sacrificing fidelity.

## 5. Ablation & Analysis

**Model Component Effectiveness.** Tab. 5 reports an ablation on cell classification following the settings in Sec. 4.5, starting from the cell-level backbone and incrementally

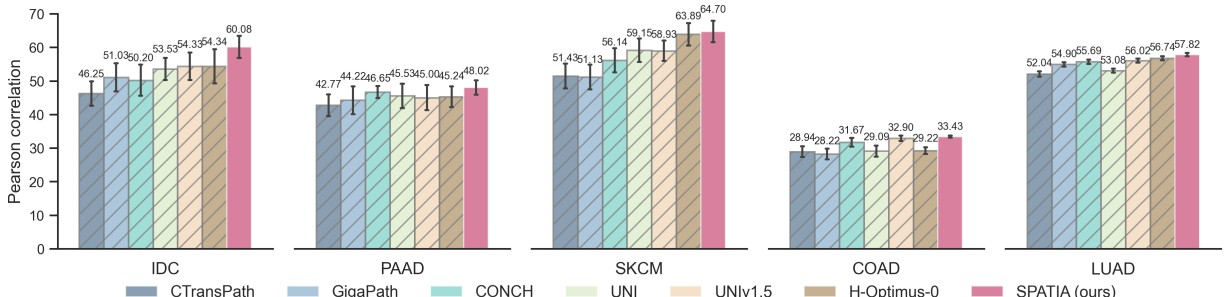

*Figure 6.* Gene expression prediction.

*Table 5.* Sub-module effectiveness evaluation of SPATIA

| Method | Loss (↓) | Accuracy (↑) |
|---|---|---|
| Cell level only | 0.405 | 0.93 |
| + MAE loss | 0.396 | 0.94 |
| + Multi-level | 0.369 | 0.97 |
| + Fusion | 0.361 | 0.98 |

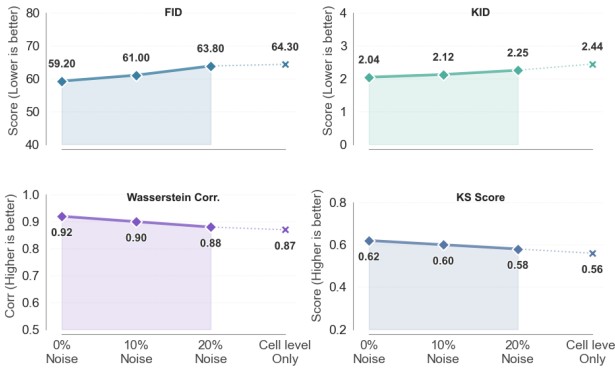

*Figure 7.* Robustness evaluation of SPATIA in OT-based control-perturbed matching.

adding: (i) reconstruction loss, (ii) multi-level hierarchy, (iii) cross-attention fusion. Adding the multi-level hierarchy produces the largest single improvement, underscoring the value of aggregation across levels.

Collectively, these results show that each component meaningfully enhances our multi-level representation. Removing niche-level context also leads to consistent degradation in both image fidelity and morphology correctness. Specifically, FID increases from 59.2 to 60.3 and KID from 2.04 to 2.24, showing poorer visual realism. Additionally, we test the cell-only variant of SPATIA to directly test whether the conditional flow is exploiting dataset co-occurrence rather than genuine spatial conditioning. This ablation isolates the effect of spatial context while keeping architectural capacity comparable. The results are shown in Fig. 7.

**OT Robustness Analysis.** To assess the robustness of our model to imperfect OT-based control-perturbed matching, we conducted a pairing-noise ablation. While our OT procedure incorporates lineage consistency and a spatial-adjacency penalty to discourage implausible matches, the flow model itself is designed to learn distributional perturbation directions rather than exact one-to-one trajectories. We therefore randomly corrupted 10-20% of OT pairs by swapping perturbed targets within the same slide and re-trained the flow module under identical settings. In Fig. 7, performance degrades steadily with increasing noise (e.g., FID: 59.2 → 61.0 → 63.8), confirming that SPATIA remains stable under moderate pairing errors and does not rely on brittle correspondences during conditional generation.

We further test stronger corruption settings, including 30

~ 40% within-slide corruption and 20% cross-slide corruption in Tab. 12. SPATIA degrades gracefully and remains competitive with baselines under substantial pairing noise.

## 6. Conclusion

We presented SPATIA, a unified model for representing and generating spatial cell phenotypes. SPATIA links cellular morphology, transcriptomics, and spatial organization through a hierarchical architecture that models information from single cells to local niches and whole tissue context. To model morphological perturbations without paired before-and-after observations, SPATIA uses entropy-regularized optimal transport to construct weak control-target pairs and confidence-aware flow matching to reduce the influence of uncertain matches. Morphology-profile alignment further constrains generated cells to match target-state phenotypic feature distributions.

Trained on MIST, an atlas of 25.9M cells from 74 datasets, SPATIA improves generative fidelity by 8% and predictive performance by up to 3% across 12 benchmarks. These results show that joint modeling of morphology, gene expression, and spatial context can support both prediction and controllable generation in spatial transcriptomics.

## Impact Statement

This paper presents methods for modeling and generating cellular morphology conditioned on spatial transcriptomic context, with the aim of advancing machine learning tools for computational biology and biomedical research. Potential positive impacts include supporting hypothesis generation, improving analysis of cellular state transitions, and enabling in-silico exploration of phenotypic variation when experimental measurements are limited.

The proposed approach operates on de-identified cellular imaging and gene expression data collected for research purposes and is not designed for direct clinical use or decision-making. As with many data-driven models, performance may be influenced by dataset composition, platform-specific characteristics, and annotation quality, which could affect generalization across biological settings. Model outputs should therefore be interpreted as predictive simulations rather than experimentally validated outcomes.

Overall, this work focuses on methodological advances in machine learning and their application to biological data analysis. We do not identify specific ethical or societal risks beyond those commonly associated with computational modeling in biomedical research.

## Acknowledgements

We appreciate the valuable discussions with Yepeng Huang on task design, Shanghua Gao and Soumya Ghosh on model design, and Changdi Yang on image generation design. We also thank Boyang Fu and Walker Rickord for their helpful feedback on the manuscript. We gratefully acknowledge the support by NSF CAREER Award 2339524, ARPA-H Biomedical Data Fabric (BDF) Toolbox Program, Amazon Faculty Research, Google Research Scholar Program, AstraZeneca Research, GlaxoSmithKline Award, Roche Alliance with Distinguished Scientists (ROADS) Program, Sanofi iDEA-iTECH Award, Boehringer Ingelheim Award, Merck Award, Optum AI Research Collaboration Award, Pfizer Research, Gates Foundation (INV-079038), Chan Zuckerberg Initiative, Collaborative Center for XDP at Massachusetts General Hospital, John and Virginia Kaneb Fellowship at Harvard Medical School, Biswas Computational Biology Initiative in partnership with the Milken Institute, Harvard Medical School Deans Innovation Fund for the Use of Artificial Intelligence, and the Kempner Institute for the Study of Natural and Artificial Intelligence at Harvard University. Any opinions, findings, conclusions or recommendations expressed in this material are those of the authors and do not necessarily reflect the views of the funders.

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

# A. Training & Implementation Details

**Training Details.** The hierarchical modules ($\mathcal{F}_{cell}$, $\mathcal{F}_{niche}$, $\mathcal{F}_{tissue}$) are trained concurrently with the corresponding dataset using primary reconstruction losses (MAE) computed on the corresponding embeddings ($\mathbf{z}_i^c$, $\mathbf{z}_i^n$, $\mathbf{z}_i^t$).

We adopt a hierarchical and localized batching strategy, which keeps sequence lengths bounded and independent of the total number of cells in a slide. The cell encoder is pretrained independently using individual (image, expression) pairs. A training batch contains a fixed B number of sampled cells, and attention is computed only within this batch, not across the full slide. For each sampled cell, its $256\times256$ px niche is extracted and encoded as one niche token (10-30 neighboring cells). The niche encoder is pretrained separately and does not process the entire tissue at once. The resulting complexity for three levels is $O((3B)^2)$, ensuring feasibility regardless of slide size. Pretraining each level individually and fine-tuning jointly over localized patches avoids any quadratic explosion and makes SPATIA scalable to slides with hundreds of thousands of cells.

We use the AdamW optimizer (Kingma & Ba, 2017) with a learning rate of 1e-3. Regarding downstream tasks, we follow the settings from CellPLM and HEST-1k. Specifically, For Biomarker Status Prediction Tasks, we fine-tune the image encoder and train the expression encoder from scratch. We use a base learning rate of $10^{-4}$ for the image encoder and $10^{-3}$ gene expression encoder. Only the last 3 layers of the model were fine-tuned, with a layer-wise learning decay rate of 0.7. For Gene Expression Prediction Tasks, we utilize an XGBoost regression model with 100 estimators and a maximum depth of 3. We evaluate 3-4 seeds, and standard deviation is $\pm$ 0.05

**Model Architecture.** Our model architecture, based on `scPrint`, has core components including: a `GeneEncoder` for processing gene expression data, which contains an embedding layer (`Embedding`) and a continuous value encoder (`ContinuousValueEncoder`); an image processing module based on `ViTMAEForPreTraining`, comprising a 12-layer ViT encoder (`ViTMAEEncoder`) and an 8-layer ViT decoder (`ViTMAEDecoder`); and an 8-layer `FlashTransformerEncoder` as the main sequence transformer.

*Table 6.* Model Hyperparameters for SPATIA

| Component | Parameter | Value |
|---|---|---|
| *Gene Processing Module* | | |
| Gene Encoder | Embedding Dimension | 256 |
| | Vocabulary Size (Genes) | 23122 |
| Expression Encoder | Output Dimension | 256 |
| | Dropout | 0.1 |
| *Core Transformer* | | |
| Flash Transformer Enc. | Number of Blocks | 8 |
| | Hidden Size (d_model) | 256 |
| | MLP Intermediate Size | 1024 |
| | Dropout | 0.1 |
| *Image Processing Module (ViTMAE)* | | |
| ViT Encoder | Hidden Size | 768 |
| | Number of Layers | 12 |
| | Patch Size | 16x16 |
| | MLP Intermediate Size | 3072 |
| ViT Decoder | Hidden Size | 512 |
| | Number of Layers | 8 |
| | MLP Intermediate Size | 2048 |
| *Fusion Layers* | | |
| Image Fusion Layer | Dimension | 768 |
| | Dropout | 0.1 |
| Expression Fusion Layer | Dimension | 256 |
| | Dropout | 0.1 |
| *Output Decoders* | | |
| Expression Decoder | Hidden Dimension | 256 |
| | Dropout | 0.1 |

The model also integrates multiple `FusionLayers` for multimodal feature fusion, an `ExprDecoder` for gene expression reconstruction, and multiple `ClsDecoders` for downstream classification tasks. Key hyperparameters are summarized in Tab. 6.

Pretrained weights are essential for SPATIAs performance. The scPRINT gene encoder is pretrained on millions of scRNA-seq cells and is specifically designed to denoise expression, correct batch effects, and infer genegene interactions; training a gene encoder of similar scale from scratch on MIST is not feasible and leads to substantial performance degradation. Likewise, the pretrained ViT image encoder provides strong morphology priors that significantly improve single-cell feature quality. To quantify this, we compared SPATIA with (i) pretrained vision encoders and (ii) the same architectures trained from scratch (random initialization), while keeping the gene encoder fixed (as scPRINT is currently one of the few large pretrained models for scRNA-seq).

**Design Selection.** In niche level, the expression vectors are summed across cells. The summation operation is only applied

at the niche level to obtain a coarse regional representation, similar to how pseudo-spots are constructed by aggregating single-cell expression within fixed grid regions in prior works (Liu et al., 2023; Mason et al., 2024; Hao et al., 2024b). This aggregation is not used for any cell-level task (Tab. 3 and Tab. 4 of the manuscript). All cell-level modeling and multimodal fusion in SPATIA are performed via cross-attention, which is fully non-linear and learns context dependent relationships between morphology and gene expression features.

**Computation Analysis.** We profiled SPATIA on a full-scale training run using 4 NVIDIA H100 (80GB) GPUs for 25,000 steps. Table 7 summarizes the key system statistics. The model requires 67 GB of VRAM per device (78.7% peak utilization),

*Table 7.* Computation profile of SPATIA during full-scale training.

| Metric | Value | Notes |
|---|---|---|
| GPU Hardware | $4 \times$ NVIDIA H100 (80GB) | Full training run |
| Training Steps | 25,000 | Standard configuration |
| VRAM Usage | 67 GB/device | 78.7% peak utilization |
| GPU Utilization | 97% peak | Stable during training |
| Power Consumption | 436 W avg. | Per GPU |
| Training Time | $\sim$30 hours | Per largest checkpoint |
| Inference Time | Low latency | Suitable for deployment |

confirming that the full architecture fits comfortably within a single high-end GPU without model parallelism. During training, GPU utilization reaches 97%, with an average power draw of 436 W per device. The largest checkpoint completes in approximately 30 hours, while inference runs at low latency, making SPATIA practical for both research and downstream biological workflows.

# B. Self-Supervised Training Objectives

We train SPATIA using self-supervised objectives on paired multimodal data. The goal of this stage is to learn a unified cell embedding $\mathbf{z}_{cell}$ that jointly captures morphology and gene expression while preserving cross-modal consistency. This is achieved by reconstructing both modalities from the same embedding, forcing $\mathbf{z}_{cell}$ to retain information necessary for both image appearance and transcriptomic state.

**Image Reconstruction.** An image decoder $D_{\mathrm{img}}$ takes the unified cell embedding $\mathbf{z}_{cell}$ and reconstructs the original cell image $x$, i.e., $\hat{x} = D_{\mathrm{img}}(\mathbf{z}_{cell})$. We use a pixel-space reconstruction loss $\mathcal{L}_{\mathrm{img}} = \mathbb{E}\big[\|\hat{x} - x\|_2^2\big]$. Reconstructing morphology ensures that the embedding preserves structural and textural features (cell shape, boundaries, staining patterns). This prevents the fusion process from discarding fine-grained visual information when integrating gene signals.

**Gene Reconstruction.** A gene decoder $D_{\mathrm{gene}}$ also takes the same unified embedding $\mathbf{z}_{cell}$ and predicts the gene expression vector, $\hat{\mathbf{g}} = D_{\mathrm{gene}}(\mathbf{z}_{cell})$. We use a masked reconstruction objective $\mathcal{L}_{\mathrm{gene}} = \mathbb{E}\big[\|\hat{\mathbf{g}} - \mathbf{g}\|_2^2\big]$. Gene reconstruction forces the embedding to preserve transcriptomic variation and align with the scPRINT backbone representations. Using the same latent for both modalities enforces cross-modal consistency, meaning morphology-aware features must also explain gene expression.

**Overall Self-Supervised Objective.** The pretraining objective combines the two terms as $\mathcal{L}_{\text{self-sup}} = \lambda_{\mathrm{img}}\mathcal{L}_{\mathrm{img}} + \lambda_{\mathrm{gene}}\mathcal{L}_{\mathrm{gene}}$, where $\lambda_{\mathrm{img}}$ and $\lambda_{\mathrm{gene}}$ balance the importance of visual and transcriptomic reconstruction. This self-supervised stage learns modality-consistent cell embeddings before spatial aggregation and cond

# C. Perturbation Pairing for Generation of Spatial Transcriptomic Cell Phenotypes

## C.1. Problem Formulation

The Xenium platform provides paired imaging and gene expression for individual cells in tissue, but we cannot observe the same cell before and after a perturbation. We therefore construct spatial "pre-perturbation to post-perturbation" examples at the population level. Within each tissue sample, we pair cells from a control state with cells from a target state, enforcing lineage and spatial constraints.

Let the spatial transcriptomic dataset be $\mathcal{D} = \{(x_i, \mathbf{g}_i, s_i)\}_{i=1}^N$, where $x_i$ is the cell image (morphology), $\mathbf{g}_i$ the gene

expression vector, and $s_i \in \mathcal{S}$ denotes the discrete cell state label (e.g., cell type or functional cluster). Note that while $s$ is implicit in the main text's description of transitions, we define it explicitly here to formalize the transition logic.

Our goal is to construct weak control–target pairs $\mathcal{P}_\tau = \{(x_{ctrl}, x_{tgt}, \mathbf{g}_{ctrl}, \mathbf{g}_{tgt})\}$ for each transition type $\tau$. A transition is defined as a mapping between specific source and target states: $\tau = (s_{ctrl} \to s_{tgt})$. Each pair is interpreted as an instance of the same biological transition mechanism rather than an observation of the same physical cell before and after perturbation.

**Transition Design.** We focus on two major perturbation axes (tasks) derived from domain knowledge: The tumor progression axis ($\mathcal{T}_{tumor}$) encompasses cellular transitions that collectively model the progression from luminal ductal carcinoma in situ (DCIS) to invasive carcinoma. First, we model epithelial-mesenchymal transition (EMT) through the mapping $s^c = \text{Epi\_FOXA1}^+ \to s^t = \text{EMT-Epi1\_CEACAM6}^+$, where FOXA1-positive luminal epithelial cells transition to CEACAM6-expressing EMT-associated states that exhibit enhanced invasive potential. The immune infiltration axis ($\mathcal{T}_{immune}$) covers transitions modeling the shift from an immune-cold tumor microenvironment to an immune-hot one. T-cell activation is represented by $s^c = \text{tcm\_CD4}^+\text{T} \to s^t = \text{eff\_CD8}^+\text{T1}$, modeling the functional transition from central memory CD4+ T cells to effector CD8+ T cells, which represents a shift from immunosuppressive to cytotoxic immune responses. Angiogenesis activation follows $s^c = \text{EC\_CAVIN2}^+ \to s^t = \text{EC\_CLEC14A}^+$, capturing endothelial cell activation from CAVIN2-expressing quiescent states to CLEC14A-positive angiogenic states that facilitate immune cell infiltration and vascular remodeling within the tumor microenvironment.

**Quality Control Criteria.** We impose several biological constraints to ensure that paired transitions are valid: 1) The control and target must belong to the same developmental lineage to avoid non-biological pairings (e.g., an epithelial cell paired with an immune cell). 2) Each cell state contains at least $\theta_{min} = 50$ cells to ensure robust statistical support for the pairing. 3) We preferentially pair cells that reside in similar niches, since cellular transitions often occur within the same or adjacent spatial regions.

## C.2. Optimal Transport-Based Cell Pairing

**Expression Space Preprocessing.** For a specific transition $\tau$, we define the set of control indices $\mathcal{I}_{ctrl} = \{i : s_i = s_{ctrl}\}$ and target indices $\mathcal{I}_{tgt} = \{j : s_j = s_{tgt}\}$. To address the high dimensionality of gene expression while preserving biological signal, we apply PCA. We center each gene expression vector by the global mean $\bar{\mathbf{g}} = \frac{1}{N}\sum_{k=1}^{N} \mathbf{g}_k$ and project onto the top $d$ principal components: $\tilde{\mathbf{g}}_i = \text{PCA}_d(\mathbf{g}_i - \bar{\mathbf{g}})$. This stabilizes OT cost computation and Sinkhorn iterations.

**Sinkhorn-Knopp Algorithm.** We formulate the pairing between the control set $\{\tilde{\mathbf{g}}_i\}_{i \in \mathcal{I}_{ctrl}}$ and target set $\{\tilde{\mathbf{g}}_j\}_{j \in \mathcal{I}_{tgt}}$. We define the pairwise transport cost matrix $\mathbf{C}$, where entries are:

$$C_{ij} = \|\tilde{\mathbf{g}}_i - \tilde{\mathbf{g}}_j\|_2, \quad \text{for } i \in \mathcal{I}_{ctrl}, j \in \mathcal{I}_{tgt}. \tag{10}$$

We solve the entropy-regularized OT problem to find the optimal coupling matrix $\mathbf{P}^*$:

$$\mathbf{P}^* = \arg\min_{\mathbf{P} \in \Pi(\mu,\nu)} \langle \mathbf{P}, \mathbf{C} \rangle + \epsilon H(\mathbf{P}), \tag{11}$$

where $\Pi(\mu, \nu)$ denotes couplings between uniform source/target marginals, and $H(\mathbf{P}) = -\sum_{ij} \mathbf{P}_{ij} \log \mathbf{P}_{ij}$ is the entropy term. Entropy regularization yields a soft coupling that reflects uncertainty by distributing mass across plausible matches rather than enforcing brittle hard assignments.

Using log-domain Sinkhorn updates, the optimal coupling is:

$$\mathbf{P}^* = \exp\left((-\mathbf{C} + u^*\mathbf{1}^T + \mathbf{1}v^{*T})/\epsilon\right), \tag{12}$$

with dual potentials $u^*, v^*$. For downstream training, we derive a discrete match for each control cell $i$ by maximum coupling:

$$\pi(i) = \arg\max_{j \in \mathcal{I}_{tgt}} \mathbf{P}^*_{ij}.$$

This yields paired observations $(x_{ctrl}, x_{tgt}) = (x_i, x_{\pi(i)})$. We emphasize that this discrete pairing is a weak supervision signal derived from the soft coupling $\mathbf{P}^*$.

## C.3. Confidence-Aware OT Reweighting for Flow Matching

The OT plan $\mathbf{P}^*$ provides both a weak control–target pairing and an explicit measure of pairing uncertainty. Flow matching supervision depends on the endpoint displacement $\boldsymbol{u} = \boldsymbol{\ell}_{tgt} - \boldsymbol{\ell}_{ctrl}$; incorrect OT matches introduce biased target directions and can corrupt the learned conditional velocity field. We therefore incorporate confidence-aware reweighting.

**Confidence Score and Weighting.** Given the discrete match $\pi(i)$ for a control cell $i$, we define the OT confidence score:

$$c_i = \mathbf{P}^*_{i,\pi(i)}. \tag{13}$$

We convert $c_i$ into a normalized training weight:

$$w_i = \frac{c_i^{\alpha}}{\mathbb{E}[c^{\alpha}]}, \tag{14}$$

where $\alpha > 0$ controls emphasis on high-confidence pairs and the normalization keeps $\mathbb{E}[w] \approx 1$ for stable optimization.

**Weighted Flow Matching Objective.** Let $\boldsymbol{\ell}_{ctrl} = \mathrm{Enc}(x_i)$ and $\boldsymbol{\ell}_{tgt} = \mathrm{Enc}(x_{\pi(i)})$ be the latent representations of the matched pair. We sample the bridge time $\lambda \sim \mathcal{U}(0, 1)$ and define $\boldsymbol{\ell}_\lambda = (1 - \lambda)\boldsymbol{\ell}_{ctrl} + \lambda\boldsymbol{\ell}_{tgt}$. Given condition embedding $\mathbf{z}_{cond}$ (defined below), the confidence-weighted flow matching loss is:

$$\mathcal{L}^w_{FM} = \mathbb{E}_{\lambda,i}\left[w_i \left\|v_\theta(\boldsymbol{\ell}_\lambda, \lambda \mid \mathbf{z}_{cond}) - (\boldsymbol{\ell}_{tgt} - \boldsymbol{\ell}_{ctrl})\right\|_2^2\right]. \tag{15}$$

This objective learns a conditional velocity field consistent with the linear bridge distribution while reducing the impact of uncertain OT supervision via $w_i$.

**Condition-Contrastive Regularization.** To improve condition identifiability without pulling toward incorrect endpoint directions, we contrast the true condition with an incorrect transition condition. We form an incorrect condition embedding $\mathbf{z}^-_{cond}$ by replacing the transition descriptor $\tau$ with $\tau^- \neq \tau$ while keeping the same control context. We enforce:

$$\mathcal{L}_{cond} = \mathbb{E}_\lambda\left[\|v_\theta(\boldsymbol{\ell}_\lambda, \lambda \mid \mathbf{z}_{cond}) - \boldsymbol{u}\|_2^2 - \|v_\theta(\boldsymbol{\ell}_\lambda, \lambda \mid \mathbf{z}^-_{cond}) - \boldsymbol{u}\|_2^2\right]_+, \tag{16}$$

where $[\cdot]_+ = \max(\cdot, 0)$. This ensures the true transition explains the observed displacement better than a random transition.

## C.4. Dataset Construction Pipeline

The complete dataset construction process is formalized in Algorithm 1, which integrates biological constraints, optimal transport theory, and quality control measures to generate biologically meaningful perturbation pairs.

## C.5. Experimental Design

**Dataset Characteristics.** Our methodology was applied to the Xenium breast cancer spatial transcriptomics dataset, which provides comprehensive single-cell resolution data with matched morphological information. The dataset contains 165,423 individual cells profiled across 70,611 genes, with 48 distinct cell state annotations derived from expert curation. Each cell is associated with high-resolution H&E histology images that capture morphological features at single-cell resolution, enabling direct correlation between transcriptional states and cellular morphology.

**Generated Perturbation Pairs.** The biologically-informed pairing pipeline successfully generated 1,584 perturbation pairs across two primary biological tasks. Task 1 (Tumor Progression) yielded 798 pairs distributed across three biological processes: EMT transition (266 pairs), proliferation activation (266 pairs), and lineage conversion (266 pairs). Task 2 (Immune Infiltration) produced 786 pairs spanning two processes: T-cell activation (400 pairs) and angiogenesis activation (386 pairs). This distribution reflects both the natural abundance of different cell states in the breast cancer tissue and our balanced sampling strategy to ensure sufficient statistical power for each transition type while maintaining biological authenticity. For example, a pairing result may look like

{x_ctrl_id, x_tgt_id, state_A, state_B, cell_type, niche_ctrl, niche_tgt, transition_tag, task_name, patient_id, slide_id, spatial_distance_um, match_score}:

{100119,131051, Epi_FOXA1+, EMT-Epi1_CEACAM6+, Epithelial, Epi-Immune, EMT-Immune, EMT_transition, tumor_progression, P001, S07, 38, 0.87}

**Algorithm 1** Biologically-Informed Perturbation Pairing & Signature Computation

---

1: **Require:** Dataset $\mathcal{D} = \{(x_k, \mathbf{g}_k, s_k)\}$, transition axes $\mathcal{T}$, threshold $\theta_{\min}$, regularization $\epsilon$, PCA dim $d$
2: **Ensure:** Paired dataset $\mathcal{P}$, transition signatures $\{\Delta \mathbf{g}_\tau\}$, $\{\Delta \mathbf{m}_\tau\}$
3: $\mathcal{P} \leftarrow \emptyset$
4: **for** each transition $\tau = (s_{ctrl} \rightarrow s_{tgt}) \in \mathcal{T}$ **do**
5:     $\mathcal{I}_{ctrl} \leftarrow \{ i \mid s_i = s_{ctrl} \}$
6:     $\mathcal{I}_{tgt} \leftarrow \{ j \mid s_j = s_{tgt} \}$
7:     **if** $|\mathcal{I}_{ctrl}| < \theta_{\min}$ **or** $|\mathcal{I}_{tgt}| < \theta_{\min}$ **then**
8:         **continue**
9:     **end if**
10:     $\tilde{\mathbf{G}}_{ctrl} \leftarrow \text{PCA}_d(\{\mathbf{g}_i : i \in \mathcal{I}_{ctrl}\})$                                      *// Project to latent space*
11:     $\tilde{\mathbf{G}}_{tgt} \leftarrow \text{PCA}_d(\{\mathbf{g}_j : j \in \mathcal{I}_{tgt}\})$
12:     $\mathbf{C} \leftarrow \text{ComputeCostMatrix}(\tilde{\mathbf{G}}_{ctrl}, \tilde{\mathbf{G}}_{tgt})$                      *// $C_{ij} = \|\tilde{\mathbf{g}}_i - \tilde{\mathbf{g}}_j\|_2$*
13:     $\mathbf{P}^* \leftarrow \text{Sinkhorn}(\mathbf{C}, \epsilon)$                                           *// Entropy-regularized OT*
14:     $\pi \leftarrow \text{ArgMaxCoupling}(\mathbf{P}^*)$                                 *// $\pi(i) = \arg\max_j \mathbf{P}^*_{ij}$*
15:     $\mathcal{P}_\tau \leftarrow \emptyset$
16:     **for** $i \in \mathcal{I}_{ctrl}$ **do**
17:         $j \leftarrow \pi(i)$
18:         $w_i \leftarrow \text{ComputeConfidence}(\mathbf{P}^*_{i,j})$                    *// Confidence weight*
19:         $\mathcal{P}_\tau \leftarrow \mathcal{P}_\tau \cup \{(x_i, x_j, \mathbf{g}_i, \mathbf{g}_j, w_i, \tau)\}$
20:     **end for**
21:     $\mathcal{P} \leftarrow \mathcal{P} \cup \mathcal{P}_\tau$
22:     $\Delta \mathbf{g}_\tau \leftarrow \text{Mean}(\{ \mathbf{g}_j - \mathbf{g}_i : (i, j) \in \mathcal{P}_\tau \})$                  *// Gene signature*
23:     $\Delta \mathbf{m}_\tau \leftarrow \text{Mean}(\{ \text{M}(x_j) - \text{M}(x_i) : (i, j) \in \mathcal{P}_\tau \})$       *// Morphology signature*
24: **end for**
25: **Return:** $\mathcal{P}$, $\{\Delta \mathbf{g}_\tau\}$, $\{\Delta \mathbf{m}_\tau\}$

---

## D. Further Details on Related Work

**Single Cell Models.** Foundation models for single-cell (non-spatial) transcriptomics have rapidly advanced, leveraging large-scale pretraining to support diverse downstream tasks such as cell type annotation, gene network inference, and perturbation prediction (Cui et al., 2023; Yang et al., 2022). Notable models include scGPT (Cui et al., 2023), scBERT (Yang et al., 2022), scPRINT (Kalfon et al., 2025), scMulan (Bian et al., 2024), scFoundation (Hao et al., 2024a), scInterpreter (Li et al., 2024), scHyena (Oh et al., 2023), GET (Fu et al., 2025b), SCimilarity (Heimberg et al., 2024), and xTrimoGene (Gong et al., 2023). These models are pretrained on repositories encompassing tens to hundreds of millions of cells, allowing them to capture complex transcriptional grammars, gene regulatory networks, and cellular heterogeneity across diverse biological contexts (Hao et al., 2024a; Fu et al., 2025b). However, they focus on transcriptomic data, lacking integration with spatial or imaging modalities, which are crucial for understanding cellular context within tissues.

## E. Overview of MIST Dataset

**Dataset Statistics.** Imaging-based spatial transcriptomics technologies allow us to explore spatial gene expression profiles at the cellular level. To support robust multi-scale representation learning, we introduce **MIST** (Multi-scale dataset for Image-based Spatial Transcriptomics). MIST is a large-scale, multi-platform atlas assembled from **74 distinct sources** (Janesick et al., 2023; Ren et al., 2025; Gabitto et al., 2024) spanning 17 tissue types, over 60 donors, and diverse disease states including cancer and Alzheimer's disease.

The assembled dataset integrates data from four major platforms: 10x Xenium, NanoString CosMx, BGI Stereo-seq, and 10x Visium HD. In total, MIST contains **25.9 million cells/bins** and covers over 31,000 unique genes. Crucially, MIST is designed to benchmark cross-platform and cross-disease generalization.

MIST leverages the specific imaging protocols available for each sub-collection. For the subset of data derived from Visium HD, we utilize high-resolution H&E staining images to capture tissue architecture. For the majority of the dataset,

*Table 8.* MIST Dataset Statistics: Xenium Datasets gathered from 10x Genomics (part 1).

| Collection name | Tissue | Disease | Num. cells | Num. genes |
|---|---|---|---|---|
| Xenium_Preview_Human_Lung_Cancer_With_Add_on_2_FFPE | lung | cancer | 531,165 | 392 |
| Xenium_Preview_Human_Non_diseased_Lung_With_Add_on_FFPE | lung | healthy | 295,883 | 392 |
| Xenium_Prime_Breast_Cancer_FFPE | breast | cancer | 699,110 | 5101 |
| Xenium_Prime_Cervical_Cancer_FFPE | cervical | cancer | 840,387 | 5101 |
| Xenium_Prime_Human_Lung_Cancer_FFPE | lung | cancer | 278,328 | 5001 |
| Xenium_Prime_Human_Lymph_Node_Reactive_FFPE | lymph node | reactive hyperplasia | 708,983 | 4624 |
| Xenium_Prime_Human_Ovary_FF | ovary | adenocarcinoma | 1,157,659 | 5001 |
| Xenium_Prime_Human_Prostate_FFPE | prostate | adenocarcinoma | 193,000 | 5006 |
| Xenium_Prime_Human_Skin_FFPE | skin | melanoma | 112,551 | 5006 |
| Xenium_Prime_Ovarian_Cancer_FFPE | ovary | cancer | 407,124 | 5101 |
| Xenium_V1_FFPE_Human_Brain_Alzheimers_With_Addon | brain | alzheimers | 44,955 | 354 |
| Xenium_V1_FFPE_Human_Brain_Glioblastoma_With_Addon | brain | glioblastoma | 40,887 | 319 |
| Xenium_V1_FFPE_Human_Brain_Healthy_With_Addon | brain | healthy | 24,406 | 319 |
| Xenium_V1_FFPE_Human_Breast_IDC_Big_1 | breast | invasive ductal carcinoma | 892,966 | 280 |
| Xenium_V1_FFPE_Human_Breast_IDC_Big_2 | breast | invasive ductal carcinoma | 885,523 | 280 |
| Xenium_V1_FFPE_Human_Breast_IDC_With_Addon | breast | invasive ductal carcinoma | 576,963 | 380 |
| Xenium_V1_FFPE_Human_Breast_IDC | breast | invasive ductal carcinoma | 574,852 | 280 |
| Xenium_V1_FFPE_Human_Breast_ILC_With_Addon | breast | invasive lobular carcinoma | 365,604 | 380 |
| Xenium_V1_FFPE_Human_Breast_ILC | breast | invasive lobular carcinoma | 356,746 | 280 |
| Xenium_V1_Human_Brain_GBM_FFPE | brain | glioblastoma | 816,769 | 480 |
| Xenium_V1_Human_Colorectal_Cancer_Addon_FFPE | colorectal | cancer | 388,175 | 480 |
| Xenium_V1_Human_Ductal_Adenocarcinoma_FFPE | pancreas | ductal adenocarcinoma | 235,099 | 380 |
| Xenium_V1_Human_Lung_Cancer_Addon_FFPE | lung | cancer | 161,000 | 480 |
| Xenium_V1_Human_Lung_Cancer_FFPE | lung | cancer | 278,659 | 289 |
| Xenium_V1_Human_Ovarian_Cancer_Addon_FFPE | ovary | cancer | 247,636 | 480 |
| Xenium_V1_hBoneMarrow_acute_lymphoid_leukemia_section | bone marrow | acute lymphoid leukemia | 225,906 | 477 |
| Xenium_V1_hBoneMarrow_nondiseased_section | bone marrow | healthy | 84,518 | 477 |
| Xenium_V1_hBone_nondiseased_section | bone | healthy | 33,801 | 477 |
| Xenium_V1_hColon_Cancer_Add_on_FFPE | colon | cancer | 587,115 | 425 |
| Xenium_V1_hColon_Cancer_Base_FFPE | colon | cancer | 647,524 | 325 |
| Xenium_V1_hColon_Non_diseased_Add_on_FFPE | colon | healthy | 275,822 | 425 |
| Xenium_V1_hColon_Non_diseased_Base_FFPE | colon | healthy | 270,984 | 325 |
| Xenium_V1_hHeart_nondiseased_section_FFPE | heart | healthy | 26,366 | 377 |
| Xenium_V1_hKidney_cancer_section | kidney | cancer | 56,510 | 377 |
| Xenium_V1_hKidney_nondiseased_section | kidney | healthy | 97,560 | 377 |

*Table 9.* MIST Dataset Statistics: Including Xenium, SPATCH (Multi-platform)

| Collection name | Tissue | Disease | Num. cells | Num. genes |
|---|---|---|---|---|
| Xenium_V1_hLiver_cancer_section_FFPE | liver | cancer | 162,628 | 474 |
| Xenium_V1_hLiver_nondiseased_section_FFPE | liver | healthy | 239,271 | 377 |
| Xenium_V1_hLung_cancer_section | lung | cancer | 150,365 | 377 |
| Xenium_V1_hLymphNode_nondiseased_section | lymph node | healthy | 377,985 | 377 |
| Xenium_V1_hPancreas_Cancer_Add_on_FFPE | pancreas | cancer | 190,965 | 474 |
| Xenium_V1_hPancreas_nondiseased_section | pancreas | healthy | 103,901 | 377 |
| Xenium_V1_hSkin_Melanoma_Base_FFPE | skin | melanoma | 106,980 | 282 |
| Xenium_V1_hSkin_nondiseased_section_1_FFPE | skin | healthy | 68,476 | 377 |
| Xenium_V1_hSkin_nondiseased_section_2_FFPE | skin | healthy | 90,106 | 377 |
| Xenium_V1_hTonsil_follicular_lymphoid_hyperplasia | tonsil | hyperplasia | 864,388 | 377 |
| Xenium_V1_hTonsil_reactive_follicular_hyperplasia | tonsil | hyperplasia | 1,349,620 | 377 |
| Xenium_V1_humanLung_Cancer_FFPE | lung | cancer | 162,254 | 377 |
| Xenium_V1_human_Pancreas_FFPE | pancreas | cancer | 140,702 | 377 |
| Xeniumranger_V1_hSkin_Melanoma_Add_on_FFPE | skin | melanoma | 87,499 | 382 |
| SEA-AD Dataset | brain | Alzheimer's | 1,541,477 | 480 |
| SPATCH_Stereo-seq_v1.3 | OV/HCC/COAD | cancer | 1,952,831 | ∼31k |
| SPATCH_Visium_HD_(FFPE) | OV/HCC/COAD | cancer | 1,563,567 | ∼18k |
| SPATCH_Visium_HD_(FF) | OV/HCC/COAD | cancer | 1,582,310 | ∼17k |
| SPATCH_Xenium_5K | OV/HCC/COAD | cancer | 977,299 | 5,001 |
| SPATCH_CosMx_6K | OV/HCC/COAD | cancer | 730,656 | 6,175 |

including the Xenium and CosMx collections, we utilize DAPI nuclear staining images, providing precise localization for cell segmentation and morphological embedding. Dataset statistics are detailed in Tab. 8 and Tab. 9.

**Data Processing.** We address varying cell sizes by first computing a bounding box for each cell and determining a global scale factor from the largest bounding box in the slide. All cells are resized using this single scale, which preserves biologically meaningful variation in absolute cell size. For each cell, the cropped patch is resized with the global scale and then padded to 256×256, ensuring a fixed input dimension while keeping only that cell in the image. Padding prevents pixels from neighboring cells, which correspond to different expression vectors from being incorporated, avoiding modality mismatch. Additionally, Xenium provides high-quality cell contours, which we retain to preserve exact spatial size information even after resizing and padding.

In MIST, niches are defined using a non-overlapping 256×256 px fixed grid applied uniformly across the slide (Xenium resolution: 0.2125 m/px). All cells whose centroids fall within a grid tile are grouped into the same niche. For each niche, we aggregate the gene expression vectors of its constituent cells (using the pooled representation described earlier) and extract the corresponding regional image patch. This choice follows widely adopted patch-based strategies in spatial transcriptomics and computational pathology (Navidi et al., 2025; Huang et al., 2025b; Fu et al., 2025a; Huang et al., 2025b). We also empirically validated that the chosen niche size is biologically reasonable. A 256×256 px region typically contains around 10-30 cells, depending on tissue density, which aligns with common definitions of microenvironments such as tumor margins, lymphocytic aggregates, and stromal niches in pathology. We visualize this distribution in Fig. 2. At the tissue level, we group 4×4 neighboring niches into a 1024×1024 px region, enabling the model to capture coarse-scale patterns such as tumor invasion fronts and broad architectural organization. This multi-level design allows SPATIA to model both local neighborhood interactions and larger-scale spatial structure.

**Batch Effect Discussion.** To assess potential batch effects in the MIST atlas, we analyzed all Xenium datasets by constructing a common gene space, sampling 2,000 cells from each dataset, and performing joint PCA followed by UMAP. Silhouette scores computed on the PCA embeddings show low cluster separation by donor, and UMAP visualizations suggest that cells organize primarily by biological identity rather than by dataset source. Results are shown in Fig. 5. These findings suggest that, within the Xenium subset and under our preprocessing pipeline, technical batch variation is modest relative to biological variation.

**Effect of Batch-Effect Mitigation.** Fig. 8 visualizes the raw or PCA-level embeddings before representation-level mitigation, where clear dataset-driven clustering can be observed across datasets. After applying SPATIA's normalization, multimodal

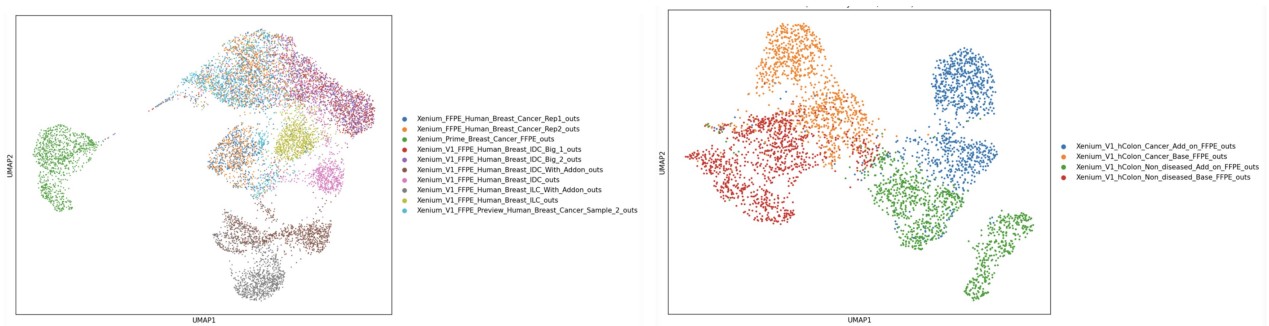

*Figure 8.* **Embeddings before batch-effect mitigation.** UMAP visualization of PCA-level embeddings prior to SPATIA normalization, multimodal pretraining, and batch-invariant representation learning. Cells cluster primarily by dataset/source, indicating dataset-driven variation before mitigation. In contrast, Fig. 5 shows the corresponding embeddings after SPATIA, where batch separation is reduced while biological structure is preserved..

pretraining, and batch-invariant representation learning strategy, Fig. 5 shows reduced batch separation while preserving biological structure. This suggests that SPATIA mitigates technical variation without collapsing biologically meaningful cell-state organization.

Our design follows recent spatial-omics foundation models such as scGPT-spatial, which combine principled normalization with large-scale pretraining to mitigate cross-slide and cross-platform variation. We also note that aggressive batch correction can distort spatial domains and remove biologically meaningful variation; therefore, SPATIA relies on normalization, donor-disjoint evaluation, and multimodal pretraining rather than applying bespoke correction to each dataset.

**Potential Information Leakage Discussion.** Since cell-level embeddings attend to niche/tissue features, there is a risk of information leakage across scales. To prevent this, our pretraining is entirely self-supervised and contains no perturbation labels or signatures, so no nicheperturbation leakage can occur. Most downstream tasks are single-level and do not combine niche-level perturbation signals with cell-level labels. For tasks where both cell and niche representations are used, the model receives both modalities as explicit inputs, not as labels, so niche correlations do not generate shortcut pathways.

Additionally, all evaluations use donor-disjoint splits, meaning that all modalities (morphology, expression, spatial context) from a donor appear exclusively in either the training set or the test set. Because donor identity is the dominant source of morphological variation, this prevents tissue-level morphology from leaking into the prediction task. Moreover, the performance gains persist even when using single-modality ablations (only morphology or only expression), confirming that improvements are driven by the learned multimodal representations rather than unintended cross-slide or cross-donor leakage.

## F. Additional Experiments and Ablation Analyses

**Scaling Evaluation.** To analyze whether SPATIA benefits from larger backbone capacity, we evaluated multiple Vision Transformer (ViT) variants while keeping the gene-expression encoder fixed (as scPRINT is currently among the few pretrained models for scRNA-seq). We compared a Base and a Large encoder variant.

*Table 10.* Scaling behavior of ViT-based image encoders

| Vision Encoder | Params (M) | Loss | | Clustering | |
|---|---|---|---|---|---|
| | | Train ↓ | Val ↓ | ARI ↑ | NMI ↑ |
| SPATIA-ViT-Base | 86M | 0.4620 | 0.4518 | 0.870 | 0.831 |
| SPATIA-ViT-Large | 307M | 0.4885 | 0.4637 | 0.842 | 0.805 |

Interestingly, the larger ViT model performs worse than the Base variant. We attribute this decline to two factors: (1) natural-image priors inherited by larger ViTs transfer poorly to fluorescence-based morphology data, and (2) despite dataset scale, spatial single-cell assay variability remains limited relative to natural-image corpora, leading to overfitting of fine-grained noise rather than meaningful structure.

**Importance of Pretraining for Image Encoder Initialization.** We initialize the ViT-MAE image encoder from ImageNet-1k pretrained weights and further adapt it on morphology patches during SPATIA training. To quantify the effect of pretrained initialization, we trained SPATIA with a randomly initialized vision encoder and compared it against a version using pretrained weights on morphology patches.

*Table 11.* Effect of pretrained weights on SPATIA performance

| **Vision Encoder Init.** | **Loss** | | **Clustering** | |
|---|---|---|---|---|
| | Train ↓ | Val ↓ | ARI ↑ | NMI ↑ |
| SPATIA (from scratch) | 0.4838 | 0.4625 | 0.813 | 0.774 |
| SPATIA (from pretrained) | 0.4620 | 0.4518 | 0.870 | 0.831 |

Pretrained initialization substantially improves optimization stability, convergence, and downstream clustering quality. This suggests that incorporating priors from morphology-aware pretraining is crucial for reliable representation learning under limited morphological variation.

**Robustness to Weak Pairing Noise.** To evaluate robustness to noisy weak pairing, we corrupt OT-derived control–target pairs under both within-slide and cross-slide settings. Within-slide corruption randomly replaces a fraction of target matches with incorrect targets from the same slide, while cross-slide corruption swaps targets across slides from the same donor, creating a more challenging mismatch pattern that better approximates real pairing errors across tissue regions.

*Table 12.* **Robustness under OT pairing corruption.** SPATIA degrades gracefully under stronger weak-pairing noise. Cross-slide corruption is more challenging than within-slide corruption, consistent with the increased heterogeneity of mismatched tissue regions.

| Corruption | FID ↓ | KID ↓ | Wass. Corr. ↑ | KS ↑ |
|---|---|---|---|---|
| 0% baseline | 59.1 | 2.04 | 0.93 | 0.62 |
| 30% within-slide | 66.1 | 2.38 | 0.84 | 0.55 |
| 40% within-slide | 69.4 | 2.55 | 0.82 | 0.52 |
| 20% cross-slide | 65.2 | 2.31 | 0.87 | 0.56 |

