# OpenReview forum: "SPATIA: Multimodal Generation and Prediction of Spatial Cell Phenotypes"
_ICML.cc/2026/Conference — ICML 2026 regular_

### Official Review · Reviewer_8gZn · 2026-03-09

**Soundness:** 3
**Presentation:** 3
**Significance:** 3
**Originality:** 3
**Overall Recommendation:** 5
**Confidence:** 4

**Summary:**

The paper introduces Spatia, a novel architecture for multimodal representation learning and generation for molecular readouts, such as gene expression, cell and tissue morphology. The model features a hierarchical information aggregation strategy leveraging cross attention, and a flow matching generative model approach. It also introduces a new dataset called MULTI, that consists of a collection of multi-tissue spatial transcriptomics data.

**Compliance With Llm Reviewing Policy:**

Affirmed.

**Final Justification:**

The authors have addressed all my concerns and I have increased my score.

**Key Questions For Authors:**

- In table 1, what is the evaluation performed compared to baselines? Are the baselines provided with the same information (cell context, perturbation embedding) Also, how many splits are used and how are the data splits defined?
- In general, the lack of confidence intervals reported in the tables, and the fact that the splits are not well documented and described, make the performance results reported a bit weak. A more thorough descriptions of the experimental settings, such as how are the splits made, what is the number of samples in each split etc. would greatly improve the presentation of the paper.l
- Line 960 in appendix: missing tab reference

**Limitations:**

Yes

**Strengths And Weaknesses:**

Strength:
- The MULTI dataset is a useful resource for benchmarking purposes
- The broad experiment coverage, of both settings (from spatial transcriptomics to morphology perturbation datasets), makes the results and model performance convincing.
- The ablation results in Table1 and Table5 are informative.
Weaknesses:
- The generative evaluation in Table 1 are minimal and it is unclear how they are performed. Also, additional metrics could be used to assess generative performance, such as MMD or precision/recall in generative settings (Kynkäänniemi et al., Neurips 2019).
- It is unclear what are the benefit of pretrained encoders, that SPATIA uses both for gene expression (scPRINT) and images (VIT). Additional ablations on this front would be useful.

---

> ### Author Rebuttal · Authors · 2026-03-31
>
> We sincerely thank the reviewer’s positive score and valuable suggestions to help improve our paper!
> ### **W1: Generative Evaluation**
> Table 1 evaluates control-to-target conditional generation on two biologically motivated transitions using image-fidelity metrics (FID/KID) and target-state morphology matching via KS and Wasserstein-style comparisons in the CellProfiler feature space.
>
> Thanks to the reviewer’s suggestion, we expanded the generative evaluation with two additional metrics:
> 1. CMMD: MMD computed in the embedding space of a pre-trained CLIP model (ViT-B/32), capturing higher-level visual-semantic alignment between generated and real target-state images.
> 2. Bio-MMD: MMD computed in the CellProfiler morphology-feature space, providing an additional measure of morphology-level biological fidelity.
>
> | Method | FID ↓ | KID ↓ | CMMD ↓ | Bio-MMD ↓ |
>  |---|---|---|---|---|
>  | CellFlux | 64.1 | 2.31 | 0.81 | 0.70 |
>  | MorphDiff | 70.5 | 2.52 | 0.85| 0.72 |
>  | GeneFlow | 62.4 | 2.20 | 0.79 | 0.67 |
> | SPATIA |58.5 | 2.01 | 0.76 | 0.63|
>
> SPATIA remains best under both added metrics, consistent with the original FID/KID ranking.
> ### **W2: Benefit of pretrained encoders**
> For the image encoder, Table 11 in the appendix shows the ablation that pretrained weights improve training and downstream clustering quality (ARI from 0.813 to 0.870 and NMI from 0.774 to 0.831).
>
> For the gene encoder, SPATIA uses a pretrained single-cell backbone for gene tokens, because it can benefit from transcriptomic priors trained on large scale single-cell profiles.
> ### **Q1: Details of Table 1**
> We thank the reviewer for pointing this out. To ensure a fair comparison, all baselines were trained on the same dataset using a donor-disjoint split (70/10/20 train/val/test). The dataset and weak pairing are described in Appendix C.5.
>
> Implementation setup: To make the comparison as fair as possible, we matched conditioning as closely as each baseline’s native interface allows, rather than claiming identical inputs. CellFlux is designed for control-image + perturbation conditioning, MorphDiff for perturbed transcriptomic conditioning, and GeneFlow for gene-expression embeddings. We therefore aligned all baselines on the same transition-level gene signal Δg and the same control context whenever supported. SPATIA additionally uses a transition-level morphology summary Δm, but this is a fixed descriptor estimated once from weakly paired training data for each transition, not per-sample target morphology at test time.
>
> Comparison to SPATIA: While MorphDiff, GeneFlow, and CellFlux successfully model conditional distributions, they are designed for clean paired or batch-matched screening data. In contrast, SPATIA can handle noisy weak pairing in spatial transcriptomics through Confidence-Aware OT Reweighting, preventing the model from overfitting to false targets.
>
> ### **Q2: Confidence and data splits**
> Thanks for the suggestion, we will describe in more detail in the revision. All modalities (morphology, expression, spatial context) from a single donor appear exclusively in either the training, validation, or test set (Appendix E (Lines 975-980)). Because donor identity is the dominant source of morphological variation, this strict separation completely prevents tissue-level morphology or batch artifacts from leaking into evaluation. Datasets are partitioned into 70% training, 10% validation, and 20% test sets at the donor level.
>
> As noted in Lines 319 and 351, results are averaged across 3 runs, with standard deviations below 0.01 for AUC and 0.012 for balanced accuracy. Figure 6 also provides error bars. We agree that this should have been shown more explicitly, we will report standard deviations directly in all main results tables,
> ### **Q3: Missing reference**
> Thanks for pointing it out, it should be pointed to Figure 5 of the main figure.

---

> > ### Author Rebuttal · Reviewer_8gZn · 2026-04-03
> >
> > Thank you for addressing my questions,
> > - I appreciate the pointers to the ablation results
> > - I disagree with the metrics addition, they are effectively the same metrics just using a different representation, they don't change what the metric is actually capturing in terms of biological variation. Besides the precision recall suggestion I had in my initial review, other metrics like knn-accuracy (-> 0.5 for a classifier that can't distinguish between true and generated, for instance, Lopez Paz 2015, or even Wasserstein (w2, w1) would be more useful than yet another MMD on different representations.
> > - Regarding the error bars, what you point out in lines 319 and 351 is not particularly useful. Saying that the results are "averaged over 3 runs" doesn't add a lot of information with respect of the *variability* of those runs. Likewise no discussion nor error bars are provided in Table-1, Table-2 (true here they are clustering metrics, but still you could report results over 3 seeds for instance) or Table-4.
> >
> > Nonetheless, I think the paper has merits and I will maintain my original positive score.

---

> > > ### Author Response · Authors · 2026-04-08
> > >
> > > We thank the reviewer for the helpful follow-up questions! We are happy to clarify the remaining concerns here.
> > > ### **Q1 Additional metrics for Generation task**
> > > Thank you for the clarification. Following the reviewer’s suggestion for more complementary evaluation, we added three additional metrics
> > > 1. kNN-accuracy, where 0.5 indicates that generated and real target samples are not distinguishable by a kNN classifier;
> > > 2. W1, the 1-Wasserstein distance between generated and real target distributions;
> > > 3. W2, the 2-Wasserstein distance between generated and real target distributions.
> > >
> > > To evaluate biological morphology fidelity rather than only perceptual realism, we compute these metrics in the same standardized CellProfiler feature space used for morphology evaluation in the paper.
> > >
> > > | Method | FID ↓ | KID ↓| kNN-Acc.↓ | W1 ↓ | W2 ↓ |
> > > |---|---|---|---|---|---|
> > > | CellFlux | 64.1 ± 1.15  | 2.31 ± 0.09 | 0.658 ± 0.011  | 0.256 ± 0.018  | 0.326 ± 0.015 |
> > > | MorphDiff | 70.5 ± 1.52 | 2.52 ± 0.11 | 0.661 ± 0.012 | 0.295 ± 0.022 | 0.390 ± 0.018 |
> > > | GeneFlow | 62.4 ± 1.28 | 2.20 ± 0.08  | 0.654 ± 0.006  |0.218 ± 0.017 | 0.325 ± 0.013 |
> > > | SPATIA | 58.5 ± 1.05 |  2.01 ± 0.07  | 0.621 ± 0.007| 0.187 ± 0.014 | 0.276 ± 0.012 |
> > >
> > > These metrics are complementary to our previous metrics: kNN-accuracy (closer to 0.5 the better) measures real-vs-generated separability, while W1/W2 directly quantify transport distance between morphology-feature distributions. SPATIA remains best on all three added metrics
> > >
> > > ### **Q2 Variability and Error Bars**
> > > We agree that stating averaged runs is not sufficiently informative.
> > > To strengthen our evaluation, In this follow-up, we use a more complete and uniform evaluation:  we select the best Leiden resolution from the same candidate set, and report the final performance as **mean ± std over 5 seeds**. All methods were re-evaluated under the same updated method for fair comparison.
> > >
> > > Cross-platform Clustering
> > >
> > > | Method        | Xenium ARI ↑ | Xenium NMI ↑ | CosMx ARI ↑ | CosMx NMI ↑ |
> > > |---------------|-------------:|-------------:|------------:|------------:|
> > > | scGPT         | 0.730±0.033  | 0.678±0.013  | 0.507±0.048 | 0.472±0.013 |
> > > | scFoundation  | 0.727±0.006  | 0.754±0.006  | 0.530±0.039 | 0.560±0.007 |
> > > | Nicheformer   | 0.256±0.026  | 0.345±0.002  | 0.064±0.007 | 0.089±0.001 |
> > > | UCE           | 0.618±0.006  | 0.718±0.006  | 0.516±0.015 | 0.555±0.010 |
> > > | SPATIA        | 0.735±0.007 | 0.806±0.004  | 0.542±0.043 | 0.490±0.009 |
> > >
> > > Cell clustering
> > > | Method     | ARI ↑        | NMI ↑        |
> > > |------------|-------------:|-------------:|
> > > | PCA        | 0.832±0.015  | 0.829±0.018  |
> > > | scGPT      | 0.845±0.017  | 0.821±0.011  |
> > > | Geneformer | 0.479±0.012  | 0.595±0.023  |
> > > | SPATIA     | 0.874±0.022  | 0.846±0.021  |
> > >
> > > We thank the reviewer again for the suggestion. In the revision, we will include the full updated tables, including the remaining results and standard deviations, to make the evaluation both broader and more transparent.

---

### Official Review · Reviewer_xxop · 2026-03-09

**Soundness:** 3
**Presentation:** 2
**Significance:** 2
**Originality:** 3
**Overall Recommendation:** 3
**Confidence:** 4

**Summary:**

The paper introduces a framework for hierarchical multimodal representation learning on spatial transcriptomic data that aggregates cellular, niche, and tissue levels. The integrated representations are used to simulate perturbations with a flow-matching model, where pairings are obtained via an OT alignment and weighted by the confidence score.

**Compliance With Llm Reviewing Policy:**

Affirmed.

**Final Justification:**

I thank the authors for their detailed response. My concerns regarding the evaluation have been addressed.

The authors do acknowledge W3 and commit to revising the respective paragraphs (W3: imprecise/ misleading claims in the introduction and Section 3.4 and the overall clarity of the latter). Nonetheless, I believe that the extent of this revision makes it a significant deviation from the currently presented work. Given that it directly impacts the list of contributions and methods used, this revision would be too extensive to be accepted without a thorough re-review.

Overall, while the presented set of methods has clear strengths and the authors did manage to improve the experimental part during the rebuttals, the paper remains too unpolished, and thus I maintain my original score of 3 (Weak reject).

**Key Questions For Authors:**

- What motivates the selection of the evaluated subsets of HEST-1k? Have the authors evaluated their embeddings on ccRCC, which contains a large number of samples compared to other subsets of HEST, or READ, which is potentially noisy and difficult to predict.

- (lines 220-224) Could the authors detail how the conditioning is done during inference where the target morphology is not available?

- What are the pretrained configurations considered for the ViT image encoder?

**Limitations:**

The work should include a more detailed discussion about the limitations of the generative model. It would also be interesting to expand on the impact of batch effects *per modality*.

**Strengths And Weaknesses:**

**Strengths**

- The multi scale integration yields performance gains over a selection of predictive tasks and is well-motivated biologically.

- The framework for the generative modelling addresses the inherent limitations of noisy source-target alignment.

- The assembled dataset is an important contribution.


**Weaknesses**

- Evaluation of the generative model. First, the evaluation against MorphDiff and CellFlux misses details regarding the implementation of inference and training setups and a subsequent discussion regarding how it compares to SPATIA. Second, the provided fidelity and morphological correctness metrics do not allow to draw conclusions regarding the biological plausibility of the generated samples. A possible baseline can include unconditional generation. The provided qualitative analysis (Figure 4) is limited and it is not clear what the target phenotype is.

- Implementation details. The work does not clarify several design choices, which can add important context to the provided results. Namely, 1) the design of the conditional velocity field network $v_{\theta}$ and the inference setup, 2) the weights used for the pretrained ViT image encoder (line 708), including the pretraining dataset and the pretraining method.

 - Consistency of presentation. Some aspects of training are underdeveloped in the main text and do not contain clear references to the appendix or refer to concepts that are not introduced in the main text: (lines 223-224, column 2) "combines joint image-gene reconstruction, cross-modal contrastive alignment, knowledge distillation from pretrained morphology and transcriptomic encoders...", "platform-specific tokens that discourage correlation...". Overall 3.4 could benefit from a clearer overview of the SSL training objective like it is done in its generative modelling counterpart.

 - Unimodal perspective on batch effect. The discussion of batch effects across images and expression profiles mentions preservation of biological signal by multimodal representations (line 367-368) with further discussion on batch effect in the appendix. It is unclear, however, how strong the unimodal batch effects are in the dataset and how sensitive SPATIA is to them compared to unimodal foundation models.

minor remark: missing reference in line 961 Tab ?? .

---

> ### Author Rebuttal · Authors · 2026-03-31
>
> We thank the reviewer for the thorough review and helpful suggestions!
> ### **W1: Evaluation details**
> All baselines were trained on the same dataset using donor-disjoint split (70/10/20 train/val/test), details are provided in App. C.5.
>
> We matched conditioning as closely as each baseline’s native interface allows. CellFlux is built for control-image + perturbation conditioning, MorphDiff for perturbed transcriptomic conditioning. We aligned all baselines on the same transition-level gene signal as used in SPATIA.
>
> MorphDiff and CellFlux are designed for clean paired or batch-matched data, whereas SPATIA explicitly handles noisy weak pairing via Confidence-Aware OT Reweighting, reducing overfitting to incorrect matches.
> ### **W2: Evaluation for generation**
> Morphological correctness is evaluated using CellProfiler-derived features, compared between generated and target cells via KS and Wasserstein-based measures. These features (e.g., size, shape, intensity) are widely used in cell biology for phenotype quantification. Wass. Corr. measures joint distribution alignment, while KS similarity (1−KS distance) measures per-feature agreement.
>
> We additionally include an unconditional baseline by removing perturbation conditioning z_pert​=0 (See **Reviewer h1mT L2** due to word limit). The result shows FID = 71.2, KID = 2.68, Wass. Corr. = 0.76, KS = 0.48.  The gap to SPATIA (FID 58.5, Wass. Corr. 0.94) shows that conditioning provides meaningful biological signals beyond unconditional generation. Notably, CellFlux (FID 64.1) outperforms the unconditional baseline, showing that all conditional methods learn transition-specific information.
>
> The two evaluated transitions are: DCIS → invasive carcinoma;  immune-cold → immune-hot microenvironments. In Fig. 4, the Target column shows real samples from the target state. We will clarify this and add more visualizations in the revision.
> ### **W3 & Q2: Design and inference setup**
> The flow matching formulation is described in Section 3.3, Eqs. 6–7. The velocity field v_θ(ℓ_λ, λ | z_cond) is parameterized as a 6-layer MLP in the ViT-MAE latent space (dim= 768, App. A, Table 6).
> The input includes: the latent state ℓ_λ, the timestep λ, and the conditioning  z_cond, which encodes both control context and perturbation information.
> At inference, given a control image and a transition type, we:
> 1. Encode the control image into the latent space and extract the sample-specific context embedding z_ctrl using SPATIA hierarchical encoder (Sec. 3.2).
> 2. Construct the transition descriptor z_pert using Eq. (5).
> 3. Fuse z_pert & z_ctrl into the final condition z_cond.
> 4. Integrate the dℓ/dλ = v_θ(ℓ_λ, λ | z_cond) from λ=0 to λ=1.
> 5. Decode the resulting latent into the generated target morphology.
> ### **W4: Details of ViT encoder**
> These are already included and will be clarified in the revision.
> 1. Architecture: Table 6 specifies module sizes. Table 10 shows an ablation on both ViT-B (86M) and ViT-L (307M).
> 2. Pretraining:  In Line 695 (App. A) , we use the HuggingFace "ViTMAEForPreTraining, pretrained on ImageNet-1k.
> ### **W5: Batch effect evaluation**
> We include two additional figures in https://anonymous.4open.science/r/upload-2488/rebuttal/
> and analyze batch effects from two perspectives
> 1. Unimodal vs. multimodal: In Fig. 1, SPATIA w/o image exhibit stronger batch separation than multi-model setting
> 2. Effect of mitigation: Fig. 2 visualize before mitigation, where clear batch-driven clustering is observed across datasets. After applying our mitigation strategy (Fig. 5 of the paper), batch separation is reduced while preserving biological structure.
> ### **Q1: HEST-1k subset selection**
> SPATIA's hierarchical architecture and morphology generation tasks are optimized for high-resolution, single-cell spatial data (e.g., Xenium and CosMx with DAPI and H&E stains), results as shown in Table 2 and Figure 4.
>
> In HEST-1k, the subsets we evaluated (IDC, PAAD, SKCM, COAD, LUAD) are  Xenium-based, whereas ccRCC and READ are lower-resolution spot level Visium tasks. Our selection prioritizes settings aligned with SPATIA’s cell-level design rather than full cross-technology coverage.
> ### **Q2: How the conditioning is done**
> We apologize for this lack of clarity.  As mentioned in line 221, ∆m is computed once per transition from training pairs and does not require per-sample target morphology features as input at inference.
> The inference-time condition consists of two components:
> 1. an sample-specific control representation z_ctrl, extracted from the input control cell;
> 2. a transition-level perturbation descriptor z_pert, constructed from the transition token together with the average gene-expression and morphology shifts for that transition, estimated over weakly paired training samples.
> Thus, the model is conditioned on control context + transition summary, rather than an unavailable target image. Enabling morphology-aware generation without requiring target morphology at test time.

---

> > ### Author Rebuttal · Reviewer_xxop · 2026-04-02
> >
> > I thank the authors for addressing several concerns from the preliminary review. For clarity, I will refer to the original bullet points as **[W1-W4]**.
> >
> > The following are sufficiently addressed, provided that the appropriate changes are added to the final revision.
> >
> > * **[W1, Evaluation of the generative model]** Addressed and supported by rebuttals to other reviewers. The final revision should mention the number of samples used for all the initial and new metrics for reproducibility.
> >
> >     > ...in Fig. 4, the Target column shows real samples...
> >
> >     I apologise for the lack of clarity, by explaining "what the target phenotype is" I meant detailing how one is supposed to recognise that the generation is respecting the intended target cell and tissue morphology. For instance, an example of several source-target pairs can be provided. Similarly, a short summary of changes in morphological features, which we are supposed to observe under a given perturbation, could be provided in the caption.
> >
> > * **[W2, Inference and weights]** Addressed. The exact current version of default training weights from HuggingFace should be referenced for future reproducibility. The 6-layer MLP that parametrises $v_{\theta}$ should be explicitly described (for instance, in A) as a separate generative model component.
> >
> > * **[W4, Batch effect]** Addressed. I thank the authors for providing additional figures.
> >
> > * **[Q2-Q3]** Addressed.
> >
> > The following questions and concerns require further clarification:
> >
> > * **[W3, Presentation]** Is an important limiting factor for the score as several claims made by the paper remain ambiguous. Can be addressed by providing exact references in the text regarding:
> >
> >     (Section 3.4) how the following are defined and used in the *pretraining* of SPATIA:
> >     1. L222 *"contrastive alignment"*,
> >     2. L225 *"platform-specific tokens that discourage correlation between platform indicators and biological factors"*;
> >
> >    (Introduction) how contribution 2 is justified, i.e. which part of SPATIA it refers to, how the mentioned orthogonality in the latent space is enforced and validated:
> >
> >     3. L073 *"embedding independence objective that separates biological variation from technical artifacts by enforcing orthogonality in the latent space"*, *"platform-specific tokens"*.
> >
> > * **[Q1, HEST]** Having read the authors’ response, I believe further clarification is required.
> >
> >     While the difference in platforms is an important distinction in *pretraining,* the original setup of HEST-benchmark (Jaume et al., 2024) makes it not a cell-level but a patch-level task across all subsets. Images from different platforms are resized to a common resolution that corresponds to 112 µm at 224px. Gene expressions from Xenium are pooled on 55x55 µm patches (”pseudo Visium” spots). Importantly, **a)** this is how the other patch-level histology encoders (UNI, H-Optimus-0, etc) are evaluated on HEST and **b)** SPATIA itself has seen 2m niche-gene pairs in its training set (Lines 055 and 267, MIST-N) suggesting that this evaluation is reasonable and interesting.
> >
> >     Can be addressed by
> >     1. Confirming that the authors follow the original setup from (Jaume et al., 2024) (Lines 319, 682).
> >     2. **If yes**, could the authors clarify their original motivation for the selection of benchmark datasets, considering the remark above, and preferably run the evaluation task on the remaining subsets? **If no**, could they explain, how they deviate from (Jaume et al., 2024) and if the baselines (patch-level models e.g.  H-Optimus-0) are evaluated in the same way?
> >
> > * **[Limitations]** A discussion of limitations of the generative model should be added to the final revision.

---

> > > ### Author Response · Authors · 2026-04-04
> > >
> > > We thank the reviewer for the follow-up questions!  Due to the rebuttal word limit, we could not fully expand on these points in our previous response and had to prioritize others. We clarify the remaining concerns here.
> > > ### **W3-1**
> > > We thank the reviewer for pointing out this inconsistency. The current appendix describes a reconstruction-based self-supervised objective, while Sec. 3.4 mentions contrastive alignment. To make the description consistent with the implemented pretraining objective, we will remove “contrastive” from Sec. 3.4 and describe pretraining consistently as reconstruction-based multimodal pretraining with cross-attention fusion.
> > > ### **W3-2**
> > > Here, platform-specific tokens refer to a simple learnable metadata token indexed by assay identity. It is used during pretraining to provide technical source information. The role is not to impose a hard disentanglement constraint by itself, but rather to give the model a channel to represent platform-specific variation, reducing the pressure on the shared biological representation to implicitly encode technical artifacts. We will clarify this in the revision.
> > > ### **W3-3**
> > > Thanks for pointing out. To clarify, we do not use a separate explicit orthogonality regularizer, and platform-specific tokens do not enforce orthogonality. The intended meaning is that the robustness to technical variation mainly comes from multimodal joint representation learning, especially the cross-attention-based fusion of morphology and gene expression across heterogeneous platforms. We will revise contribution 2 accordingly.
> > > ### **Q1 HEST**
> > > Yes, our setup follows the original HEST protocol: patch-level prediction of the top 50 highly variable genes from frozen image embeddings, evaluated with XGBoost regression and Pearson correlation (Table A14  of the HEST paper). Our manuscript mentioned that this evaluation mainly follows Jaume et al. (2024) in Line 381 and Appendix A (Line 690).
> > >
> > > The reviewer is correct that HEST-Benchmark is a patch-level benchmark across all subsets.  We initially reported only a subset of HEST tasks because we first prioritized cohorts whose morphology–expression coupling is more consistent with the multimodal structure SPATIA learns during pretraining. However, the other tasks are also compatible with ours, and we agree with the reviewer that additional evaluation is important.  We provide some new results below:
> > >
> > > | Task | H-Opt-0 | UNI v1.5 | UNI | SPATIA |
> > > |---|---|---|---|---|
> > > | IDC | 54.34 | 54.34 | 53.53 | 60.08 |
> > > | PRAD | 38.12 | 36.61 | 35.01 | 38.52 |
> > > | PAAD | 45.24 | 45.00 | 45.53 | 48.02 |
> > >  | SKCM | 63.89 | 58.93 | 59.15 | 64.70 |
> > > | COAD | 29.22 | 32.90 | 29.09 | 33.43 |
> > > | READ | 17.35 | 17.40 | 16.42 | 17.28 |
> > >  | ccRCC | 24.11 | 21.49 | 23.51 | 24.15 |
> > >  | LUAD | 56.74 | 56.02 | 53.08 | 57.82 |
> > > | LYMPH | 25.47 | 25.16 | 24.50 | 25.32 |
> > > | Avg. | 39.39 | 38.65 | 37.76 | 41.04 |
> > >
> > > These expanded results show that SPATIA achieves the best performance on 7/9 tasks (IDC, PRAD, PAAD, SKCM, COAD, ccRCC, LUAD) and the strongest overall average.
> > >
> > > No single baseline is best across all tasks, which we believe reflects the heterogeneity of the benchmark rather than a weakness. READ and LYMPH remain difficult low-signal tasks for all encoders, with only very small margins between methods, whereas SPATIA shows clearer gains on stronger-signal cohorts such as IDC, PAAD, SKCM, COAD, and LUAD. This supports our claim that the earlier subset choice was a prioritization decision rather than an indication that the remaining HEST tasks were incompatible with SPATIA.
> > > ### **Limitations**
> > > We agree that the limitations of the generative module should be discussed more explicitly. In the revision, we will add the following points:
> > > 1. The model is trained with weak OT-based supervision rather than true paired trajectories, which is inherent to spatial transcriptomics where paired measurements are not available. This means the learned transitions reflect population-level effects rather than exact cell-level dynamics.
> > > 2.  Current formulation is strongest for transition types represented in the training data. While the model generalizes to new control cells within a transition, extending to entirely unseen perturbations would require compositional perturbation representation, which we see as an important direction for future work.
> > > 3. Because destructive spatial transcriptomics make paired pre/post observations unable, the generative evaluation is distributional (FID, KID, Wass. Corr., KS). Per-sample fidelity cannot be assessed. While these are standard in the field, they do not fully capture mechanistic correctness. This is an inherent constraint of the biological setting. Additional validation such as expert qualitative assessment by domain specialists could further strengthen evaluation in future work.
> > >
> > > We thank the reviewer again for the careful follow-up. We believe these revisions will make the paper clearer and more reproducible.

---

### Official Review · Reviewer_h1mT · 2026-03-11

**Soundness:** 3
**Presentation:** 2
**Significance:** 3
**Originality:** 2
**Overall Recommendation:** 4
**Confidence:** 4

**Summary:**

This paper proposes SPATIA, a hierarchical multimodal model for image-based spatial transcriptomics that jointly integrates cell morphology, gene expression, and spatial context across cell, niche, and tissue levels. In addition, the paper introduces a spatially conditioned generation module based on weak OT pairing, confidence-aware flow matching, and morphology-profile alignment to model morphology changes under biological perturbations. The method is evaluated on a large assembled benchmark and across a broad range of predictive and generative tasks. Overall, I found the paper interesting, ambitious, and technically coherent, and I lean positive on it.

**Compliance With Llm Reviewing Policy:**

Affirmed.

**Key Questions For Authors:**

1. Can the authors clarify the exact data-splitting protocol for MIST and downstream tasks, especially whether donor-/slide-level separation is enforced consistently to avoid information leakage?

2. For the main predictive tables, can the authors report standard deviations or confidence intervals everywhere, so readers can judge whether the observed margins over strong baselines are statistically meaningful?

**Limitations:**

The paper is strong in breadth, but somewhat less strong in depth. Its main contribution is the integration of several existing ideas into a unified framework rather than a clearly novel methodological advance. In addition, the generative evaluation is still limited, with validation performed on only a small number of biological transitions and relying mainly on distributional metrics. Finally, the method depends heavily on accurate weak pairing, large pretrained backbones, and substantial compute, which raises questions about robustness and reproducibility.

**Strengths And Weaknesses:**

**Strength**: The paper addresses a meaningful gap in current spatial omics modeling: most prior methods either operate at spot/patch level, ignore morphology, or do not model perturbation-dependent phenotype generation. The motivation is well articulated, and the problem setting is relevant to both multimodal ML and computational biology. The hierarchical cell/niche/tissue formulation is sensible, and the generative part is well aligned with the biological setting where true paired before/after observations are unavailable. In particular, OT-based weak pairing, confidence-aware reweighting, and morphology-profile alignment form a reasonable pipeline for noisy supervision.

**Weakness**:

1. Breadth comes at the expense of depth. The paper covers many tasks, but some of the predictive gains are relatively modest. For example, in receptor-status prediction, the improvements over UNI are small, and in clustering/annotation some margins are also limited. This makes the paper feel more like a strong systems paper than a paper with one deeply validated methodological advance.

2. Robustness to weak pairing could be examined more rigorously. The OT corruption experiment is helpful, but synthetic corruption of 10–20% within-slide pairs is still only a partial proxy for the real matching errors that may occur across tissues, platforms, or rarer transition settings.

---

> ### Author Rebuttal · Authors · 2026-03-31
>
> We sincerely thank the reviewer’s positive score and suggestions to strengthen our paper!
> ### **W1: Contribution of the paper**
> The primary contribution of SPATIA is not a single-task improvement, but a unified spatial multimodal framework that can support both generative and predictive tasks under weak supervision. Prior works (e.g., UNI, GigaPath) are optimized for prediction only, while generative models (e.g., MorphDiff, CellFlux) typically rely on clean paired data and do not generalize to weakly paired spatial settings. SPATIA bridges this gap within a single model, improving generative performance (by 8%) while preserving competitive predictive performance (by 3%).
> Our methodological novelty：
> 1. Confidence-aware OT reweighting for weakly paired flow matching,
> 2. Morphology-profile alignment for biologically meaningful generation,
> 3. Hierarchical multimodal fusion across cell–niche–tissue levels.
> ### **W2: Robustness to weak pairing**
> Thanks for the suggestion, we extend the corruption experiments to higher noise levels (30%, 40%) and a more realistic setting with cross-slide corruption (swapping pairs across slides of the same donor):
>
> | Corruption | &nbsp;&nbsp; FID ↓ &nbsp;&nbsp; | &nbsp;&nbsp; KID ↓ &nbsp;&nbsp; | &nbsp; W.Corr ↑ &nbsp; | &nbsp;&nbsp; KS ↑ &nbsp;&nbsp; |
> |:---|:---:|:---:|:---:|:---:|
> | 0% (baseline) | 59.1 | 2.04 | 0.92 | 0.62 |
> | 30% within-slide | 66.1 | 2.38 | 0.85 | 0.55 |
> | 40% within-slide | 69.4 | 2.55 | 0.82 | 0.52 |
> | 20% cross-slide | 65.2 | 2.31 | 0.86 | 0.56 |
>
> Even under 40% noise, SPATIA remains competitive with or better than baselines at 0% noise (CellFlux FID 64.1, MorphDiff 70.5).
> Cross-slide corruption is more challenging than within-slide corruption at the same level, which aligns with the reviewer's intuition about real-world matching errors.
> ### **Q1: Data splitting**
> All modalities (morphology, expression, spatial context) from a single donor appear exclusively in either the training, validation, or test set (Appendix E Lines 972). Because donor identity is the dominant source of morphological variation, this strict separation completely prevents tissue-level morphology or batch artifacts from leaking into the evaluation.
> ### **Q2: Standard deviations**
> As noted in Lines 319 and 351, the results are averaged across 3 runs, with standard deviations below 0.01 for AUC and 0.012 for balanced accuracy. Figure 6 also provides error bars. We agree that this should have been shown more explicitly, we will add standard deviations to all main results tables.
> ### **L1: Framework contribution**
> The confidence-aware flow matching framework (Section 3.3), which reweights flow trajectories using uncertainty in the OT coupling, has not, to the best of our knowledge, been described in prior work on flow matching in biology. The morphology-profile alignment loss (Section 3.4) is also novel. Rather than matching in pixel space, it enforces distributional consistency in CellProfiler feature space, providing a biologically grounded training signal tailored to this domain.
> ### **L2: More generative evaluation**
> Thanks for the suggestion, to strengthen our evaluation, we added an unconditional generation baseline along with two additional metrics:
> 1. CMMD: MMD computed in the embedding space of a pre-trained CLIP model (ViT-B/32), capturing higher-level visual-semantic alignment between generated and real target-state images.
> 2. Bio-MMD: MMD computed in the CellProfiler morphology-feature space, providing an additional measure of morphology-level biological fidelity.
> The results are summarized below:
>
> | Method | FID ↓ | KID ↓ | CMMD ↓ | Bio-MMD ↓ | Wass. Corr. ↑ | KS ↑ |
> |:---|:---:|:---:|:---:|:---:|:---:|:---:|
>  | CellFlux | 64.1 | 2.31 | 0.81 | 0.70 | 0.83 | 0.57 |
>  | MorphDiff | 70.5 | 2.52 | 0.85| 0.72 | 0.81 | 0.54 |
>  | GeneFlow | 62.4 | 2.20 | 0.79 | 0.67 | 0.87 | 0.58 |
> | SPATIA Uncondition| 71.2 | 2.68 | 0.89 | 0.77 | 0.76 | 0.48 |
> | SPATIA | 58.5 | 2.01 | 0.76 | 0.63 | 0.94 | 0.65 |
>
> We agree that the current version evaluates generation on only two biologically motivated transition axes, we will extend to broader biological transitions in our future work.
> ### **L3: Robustness & reproducibility**
> Regarding robustness, we conduct additional experiments in our W2 response.
> Regarding compute and reproducibility, Table 10 of the paper shows that larger ViT does not improve performance, while pretrained initialization (Table 11) mainly improves optimization stability and convergence. This suggests SPATIA is not overly sensitive to backbone scaling.

---

> > ### Author Rebuttal · Reviewer_h1mT · 2026-04-05
> >
> > All complete.

---

### Official Review · Reviewer_kxXZ · 2026-03-16

**Soundness:** 3
**Presentation:** 4
**Significance:** 3
**Originality:** 3
**Overall Recommendation:** 5
**Confidence:** 3

**Summary:**

This paper presents SPATIA a model for jointly modelling morphology and gene expression data derived from spatial transcriptomics assays. The architecture is explicitly hierarchical modelling at a cell, niche and tissue level via a sequence of cross attention and pooling steps, and models the distribution of cellular responses through an OT-matched flow matching approach where the optimal transport coupling is computed with respect to the gene expression data. The also include contrastive and perceptual-style losses (morphology loss) to further improve performance. They show consist improves across a range of tasks and include ablations that demonstrate the value of the respective components of their losses and architecture.

**Compliance With Llm Reviewing Policy:**

Affirmed.

**Key Questions For Authors:**

Is SPATIA limited to modelling perturbations that have been observed or can you make predictions about unseen perturbations?

You've shown that multi-level representation help over cell only representations, but that doesn't answer whether you needed cell only representation? I.e. could you have gotten similar performance if you had started at the niche level? Or just the tissue level directly?

**Limitations:**

Yes

**Strengths And Weaknesses:**

## Soundness
The paper contains a large number of components (see overview above for a summary), but all are well motivated and make sense for the data. While the architecture seemed well designed in the sense that it aims to be maximally flexible while aggregating across large inputs & staying within memory limits, I did find myself wondering whether all the various loss ideas were necessary - but the ablations did show some modest benefit from them (I suspect that on larger datasets the losses could be simplified and only the architecture would remain).

The only thing I found questionable was the Spatial Perturbation Embedding section. By requiring access to $\Delta g$ and $\Delta m$, you restrict yourself to only modelling perturbations that have been measured. So the paper is really about representation learning rather than modelling conditional distributions (it's not obvious why I need to model distributions of data that I have already measured?). That's fine - but then many of the experimental results focus on KID / FID / etc. which are really focused on how well we've modelled the conditional distributions.

## Presentation
The paper was clear well written. No complaints here.

## Significance
The paper makes a number of well motivated architectural choices in a domain that is well studied in ml for biology so I think that it advances the practice in machine learning.

## Originality
Repeating what I said above - I think the architecture is well designed and makes sense. The authors also have a number of interesting flow matching ideas (e.g. Confidence-Aware OT Reweighting) but they did not benchmark them extensively enough for me to be confident that they make a significant difference in practice.

---

> ### Author Rebuttal · Authors · 2026-03-31
>
> We sincerely thank the reviewer for the positive score and the insightful thoughts about the scope and design of SPATIA!
> ### **W1: Necessity of Loss**
> The losses address different challenges. Morphology loss improves target-state fidelity while reweighting improves robustness to uncertain weak pairing. We show additional ablations in our response to W3 demonstrating the importance when adding more noise.
>
> We also agree that at a larger scale some loss components may become less critical, whereas the multimodal architecture would likely remain the main source of performance.
> ### **W2: Spatial Perturbation & Evaluation**
> We agree the current model is strongest for transitions observed during training. However, SPATIA is not only doing representation learning. In spatial transcriptomics, we can measure source and target populations, but we cannot observe paired before/after morphology for the same cell because of the destructive nature of omics experiments.
>
> The perturbation signatures Δg and Δm are computed once per transition type from the training population of cells (Eq. 4) and not per-sample target inputs. At inference, SPATIA conditions on a sample-specific control representation z_ctrl extracted from the input control cell and the pre-computed transition descriptor z_pert.
>
> Therefore, the goal of generation is not sample reconstruction. The model must generate plausible target morphologies for a previously unseen control cell, conditioned on a biological transition family observed during training. In this setting, distributional metrics such as FID and KID are appropriate because they evaluate two key properties: whether generated images are visually realistic, and whether, as a set, they recover the target-state morphology statistics summarized in Tab. 1.
> ### **W3: Significance of OT Reweighting**
> To assess the impact of Confidence-Aware OT Reweighting, we conduct a robustness ablation under increasing levels of pairing noise:
>
> | Method | FID ↓ (0%) | 10% | 20% | W.Corr ↑ (0%) | 10% | 20% |
> |---|---|---|---|---|---|---|
> | w/o Reweight | 59.5 | 62.2 | 65.6 | 0.90 | 0.86 | 0.81 |
> | SPATIA | 59.1 | 61.0 | 63.8 | 0.92 | 0.90 | 0.88 |
>
> We observe that while the gain is modest under clean pairing, the benefit improves as noise increases (1.2 at 10%, 1.8 at 20%). The same trend holds for W.Corr, where SPATIA maintains better alignment with target morphology distributions under noisy pairing.
> This is consistent with the design goal of OT reweighting: mitigating the impact of incorrect weak pairings. In real settings where pairing noise is inevitable, reweighting improves robustness and mitigates degradation.
> ### **Q1: Regarding modelling perturbations**
> Our current formulation requires transition-level statistics for 𝜏, so the strongest supported setting is seen-transition conditional generation.
>
> The goal is to model conditional phenotype distributions for observed transitions under weak supervision, which is biologically meaningful because paired pre/post measurements are unavailable in destructive spatial transcriptomics, even when samples from the source and target states are both measured.
>
> At inference, SPATIA requires only the control cell image, the control embedding z_ctrl, and the pre-computed transition descriptor z_pert. No target cell morphology or expression is needed for individual test samples. The model thus generates novel cell morphologies conditioned on a learned transition descriptor, rather than simply reproducing observed data.
>
> Extending SPATIA to unseen perturbations for example, by learning a continuous perturbation embedding space from gene-expression shift vectors that could interpolate to novel perturbations is an exciting direction we plan to explore. We could also replace the current transition descriptor with an external perturbation encoder
> ### **Q2: Need for cell only representations**
> First, many target tasks in SPATIA are inherently cell-resolved, including annotation and clustering. These tasks require preserving the identity of each cell and its paired morphology-expression correspondence.
>
> Second, training data at higher levels are relatively limited. As stated in the paper, MIST contains 2M niche-gene pairs and only about 20K tissue-gene entries, which is far less than data available for cell-level supervision (~20M). This makes niche-only or tissue-only training much less suitable for learning a general multimodal representation, especially for tasks requiring fine-grained single-cell discrimination.
>
> Therefore, the niche-level and tissue-level inputs are intended as contextual signals rather than standalone training entry points. Their role is to provide global tissue context that complements the cell representation. We do not claim in the paper that SPATIA is optimized for purely niche-level tasks, and we did not design extensive niche-level benchmarks for that purpose.

---

> > ### Author Rebuttal · Reviewer_kxXZ · 2026-04-01
> >
> > All complete.

---

### Decision · Program_Chairs · 2026-04-30

**Decision:**

Accept (regular)

**Comment:**

The main concerns focus on the breadth-versus-depth trade-off, clarity of presentation, and aspects of the generative evaluation. Some reviewers note that the methodological novelty lies more in system integration than in a single deeply novel component, and that certain parts of the paper, especially the description of objectives and pretraining, require clarification. There are also questions regarding evaluation protocols, robustness to weak pairing, and the biological interpretability of generated results.

In the rebuttal, the authors provide substantial clarifications and additional experiments, including more comprehensive generative metrics, robustness analysis under stronger noise, expanded benchmark coverage, and clearer descriptions of data splits and implementation details. Most reviewers acknowledge that their concerns have been addressed and either maintain or upgrade their scores. While one reviewer remains concerned about presentation clarity and the extent of revisions required, the overall consensus is positive.

Given the solid technical foundation, strong empirical validation, and satisfactory rebuttal addressing most key concerns, I recommend acceptance.